# Maternally provided LSD1/KDM1A enables the maternal-to-zygotic transition and prevents defects that manifest postnatally

Jadiel A Wasson[1,2], Ashley K Simon[3], Dexter A Myrick[1,2], Gernot Wolf[4], Shawn Driscoll[5,6], Samuel L Pfaff[5,6], Todd S Macfarlan[4]*, David J Katz[1]*

[1]Department of Cell Biology, Emory University School of Medicine, Atlanta, United States; [2]Graduate Division of Biological and Biomedical Sciences, Emory University, Atlanta, United States; [3]Department of Human Genetics, Emory University School of Medicine, Atlanta, United States; [4]The Eunice Kennedy Shriver National Institute of Child Health and Human Development, National Institutes of Health, Bethesda, United States; [5]Howard Hughes Medical Institute, The Salk Institute for Biological Studies, La Jolla, United States; [6]Gene Expression Laboratory, The Salk Institute for Biological Studies, La Jolla, United States

**Abstract** Somatic cell nuclear transfer has established that the oocyte contains maternal factors with epigenetic reprogramming capacity. Yet the identity and function of these maternal factors during the gamete to embryo transition remains poorly understood. In *C. elegans*, LSD1/KDM1A enables this transition by removing H3K4me2 and preventing the transgenerational inheritance of transcription patterns. Here we show that loss of maternal LSD1/KDM1A in mice results in embryonic arrest at the 1-2 cell stage, with arrested embryos failing to undergo the maternal-to-zygotic transition. This suggests that LSD1/KDM1A maternal reprogramming is conserved. Moreover, partial loss of maternal LSD1/KDM1A results in striking phenotypes weeks after fertilization; including perinatal lethality and abnormal behavior in surviving adults. These maternal effect hypomorphic phenotypes are associated with alterations in DNA methylation and expression at imprinted genes. These results establish a novel mammalian paradigm where defects in early epigenetic reprogramming can lead to defects that manifest later in development.

*For correspondence: todd. macfarlan@nih.gov (TSM); djkatz@ emory.edu (DJK)

**Competing interests:** The authors declare that no competing interests exist.

## Introduction

At fertilization, the epigenome of the developing zygote undergoes widespread changes in DNA methylation and histone methylation (*Aoki et al., 1997*; *Arico et al., 2011*; *Bultman et al., 2006*; *Burton and Torres-Padilla, 2010*; *Lindeman et al., 2011*; *Smith et al., 2012*; *Vastenhouw et al., 2010*; *Wu et al., 2011*). This reprogramming is driven by the deposition of maternal proteins and RNA into the zygote (*Tadros and Lipshitz, 2009*). After fertilization, zygotic genes become transcriptionally active, while the maternal transcriptional program is silenced (*Tadros and Lipshitz, 2009*). In mice, this maternal-to-zygotic transition (MZT) occurs between the one- and two-cell (1-2C) stage (*Aoki et al., 1997*; *Hamatani et al., 2004*; *Xue et al., 2013*).

Somatic cell nuclear transfer (SCNT) experiments in *Xenopus* demonstrated that the oocyte has the capacity to reprogram a differentiated somatic nucleus into a cloned embryo (*Gurdon et al., 1958*). This epigenetic reprogramming capacity of the oocyte is also conserved in mammals (*Campbell et al., 1996*). It is thought that the epigenetic reprogramming capacity of the oocyte

**eLife digest** During fertilization, an egg cell and a sperm cell combine to make a cell called a zygote that then divides many times to form an embryo. Many of the characteristics of the embryo are determined by the genes it inherits from its parents. However, not all of these genes should be "expressed" to produce their products all of the time. One way of controlling gene expression is to add a chemical group called a methyl tag to the DNA near the gene, or to one of the histone proteins that DNA wraps around.

Soon after fertilization, a process called reprogramming occurs that begins with the removal of most of the methyl tags a zygote inherited from the egg and sperm cells. The zygote's DNA is then newly methylated to activate a new pattern of gene expression. In mammals, some genes escape this reprogramming; these "imprinted" genes retain the methylation patterns inherited from the parents.

Reprogramming is assisted by "maternal factors" that are inherited from the egg cell. Once reprogramming is completed, the maternal factors are destroyed as part of a process called the maternal-to-zygotic transition. A maternal factor called KDM1A can remove specific methyl tags from certain histone proteins, but how this affects the zygote is not well understood. Now, Wasson et al. (and independently Ancelin et al.) have investigated the role that KDM1A plays in mouse development.

Wasson et al. genetically engineered mouse egg cells to contain little or no KDM1A. Zygotes created from egg cells that completely lack KDM1A die before or shortly after they have divided for the first time and fail to undergo the maternal-to-zygotic transition. Other egg cells that contain low levels of KDM1A can give rise to baby mice. However, many of these mice die soon after birth, and those that grow to adulthood behave in abnormal ways; for example, they display excessive chewing and digging. These disorders are linked to the disruption of DNA methylation at imprinted genes.

The next challenge will be to further investigate the mechanisms by which defects in maternally deposited KDM1A exert their long-range effects on imprinted genes and altered behaviour. This is particularly important because of the recent discovery of three patients with birth defects that are linked to genetic variants in KDM1A.

enables the MZT. Nevertheless, the enzymes involved in maternal epigenetic reprogramming, and the consequences of failure to reprogram, largely remain a mystery.

Recently, work in *C. elegans* implicated LSD1/SPR-5/KDM1A (previously referred to as LSD1, hereafter referred to as KDM1A) in maternal reprogramming at fertilization. Di-methylation of lysine 4 on histone H3 (H3K4me2) is acquired in gamete genes during the specification and maintenance of the germline. The H3K4me2 demethylase KDM1A is maternally deposited into the zygote at fertilization and required to prevent H3K4me2 from being inherited transgenerationally. Without KDM1A, H3K4me2 accumulates at gamete genes across generations and results in the inappropriate expression of these genes. This correlates with increasing sterility in the population over time (*Katz et al., 2009*).

The requirement to reprogram H3K4 methylation is also suggested to be a critical step in SCNT. During SCNT in *Xenopus*, the cloned embryos often inappropriately express genes from the tissue where the somatic nucleus was derived (*Ng and Gurdon, 2005*). This transcriptional memory can be eliminated by overexpressing histone H3 with a mutated K4 residue (*Ng and Gurdon, 2008*). This work implies that H3K4 methylation may be the carrier of transgenerational transcriptional memory, and that complete reprogramming may require the removal of this modification.

Genomic imprinting is the monoallelic expression of a small number of genes based on their parent of origin (*Bartolomei and Tilghman, 1997*). Imprinted genes are dependent upon DNA methylation at small imprinting control regions (ICRs) associated with these loci (*Bartolomei and Tilghman, 1997*). At ICRs, CpG methylation is established in the gametes and maintained throughout the development of the offspring (*Bartolomei, 2009*). In mammals, there are two amine-oxidase histone demethylases, LSD2/KDM1B (hereafter referred to as KDM1B) and KDM1A (*Ciccone et al., 2009*;

*Shi et al., 2004*). KDM1B is expressed mainly in the oocyte and required for the establishment of maternal imprints at imprinted loci (*Ciccone et al., 2009*; *Stewart et al., 2015*). Without KDM1B, embryos derived from these oocytes exhibit a maternal effect embryonic lethality phenotype prior to mid-gestation (*Ciccone et al., 2009*). This demonstrates that, similar to *C. elegans*, amine oxidase-type histone demethylases can function maternally in mammals.

KDM1A is a component of several protein complexes. As part of the CoREST complex, it specifically demethylates H3K4me1/2, but not H3K4me3 (*Shi et al., 2004*; *You et al., 2001*). Alternatively, when associated with the Androgen Receptor complex KDM1A has been shown to demethylate H3K9me2 in vitro (*Metzger et al., 2005*). KDM1A is an essential gene in mammalian development, as homozygous mutants fail to develop properly after implantation and die prior to embryonic day 8 (e8) (*Foster et al., 2010*; *Macfarlan et al., 2011*; *Wang et al., 2007*). To determine whether KDM1A may also be involved in maternal reprogramming, we conditionally deleted *Lsd1/Kdm1a* (previously referred to as *Lsd1*, hereafter referred to as *Kdm1a*) in mouse oocytes with three different maternal *Cre* transgenes. Deletion of *Kdm1a* maternally with either *Zp3-Cre* or *Gdf9-Cre* results in embryonic lethality primarily at the 1-2C stage and these embryos fail to undergo the MZT. This suggests that KDM1A plays a conserved role in maternal reprogramming. Surprisingly, deletion of *Kdm1a* maternally with *Vasa-Cre* results in an incomplete loss of KDM1A in oocytes. This uncovers a hypomorphic effect in which some surviving animals exhibit long-range developmental defects, including perinatal lethality and behavioral abnormalities. These defects are associated with disruption of the epigenetic landscape, including aberrant DNA methylation and expression at imprinted loci. These results demonstrate that defects in epigenetic reprogramming between generations can lead to abnormalities later in development.

## Results

### KDM1A is expressed throughout oocyte development

Based on the previously demonstrated maternal role of KDM1A in *C. elegans* and KDM1B in mice, we first asked if KDM1A is expressed in mouse oocytes (*Ciccone et al., 2009*; *Katz et al., 2009*). RNA-seq datasets from ovulated oocytes and 2C-stage embryos suggested abundant KDM1A transcripts in these cells (*Macfarlan et al., 2012*). Immunofluorescence (IF) and immunohistochemistry (IHC) with an antibody raised against KDM1A confirms that KDM1A is expressed in the oocyte nucleus and in the surrounding follicle cells throughout oocyte development (*Figure 1A*, *Figure 1—figure supplement 1A–L*). Therefore, to determine if KDM1A functions in oocytes, we conditionally deleted *Kdm1a* by crossing *Kdm1a^{fl/fl}* mice (*Wang et al., 2007*) to three different maternal *Cre* transgenic lines, *Ddx4/Vasa-Cre* (*Gallardo et al., 2007*), *Gdf9-Cre* (*Lan et al., 2004*) and *Zp3-Cre* (*de Vries et al., 2000*) (oocytes and embryos from *Kdm1a^{fl/Δ}::Ddx4/Vasa^{Cre}*, *Kdm1a^{fl/Δ}::Gdf9-Cre*, and *Kdm1a^{fl/Δ}::Zp3-Cre* mice will be referred to hereafter as *Kdm1a^{Vasa}*, *Kdm1a^{Gdf9}* and *Kdm1a^{Zp3}* respectively). *Vasa-Cre* is expressed in the germline beginning at e18 and induces full deletion by birth (*Figure 1B*) (*Gallardo et al., 2007*). *Gdf9-Cre* is expressed in oocytes beginning at postnatal day 3 (P3), including in primordial follicles (*Lan et al., 2004*), while *Zp3-Cre* is also expressed in oocytes, but beginning at P5 in primary follicles (*Figure 1B*) (*de Vries et al., 2000*). KDM1A IHC and IF demonstrate that deletion of *Kdm1a* with either *Gdf9-Cre* or *Zp3-Cre* results in the complete loss of KDM1A from the oocyte nucleus (*Figure 1C–F,I*). Crossing the *Kdm1a^{fl/fl}* allele to *Vasa-Cre* also results in complete maternal deletion, as can be determined by the 100% segregation of the maternally deleted allele to the offspring (all of offspring from *Kdm1a^{Vasa}* mothers crossed to wild-type fathers result in heterozygous -/+ offspring) (*Figure 1—figure supplement 2*). This demonstrates that, similar to deletion via *Gdf9-Cre* or *Zp3-Cre*, crossing the *Kdm1a^{fl/fl}* allele to *Vasa-Cre* also effectively deletes *Kdm1a* maternally. However, for reasons that are not clear, deletion of *Kdm1a* with *Vasa-Cre* results in a hypomorphic effect, where 33.3% of oocytes completely lack KDM1A protein, but 66.7% retain a low level of KDM1A (*Figure 1G–I*). This incomplete effect is surprising because the *Vasa-Cre* transgene is reported to be expressed earlier in the germline than either *Gdf9-Cre* or *Zp3-Cre* (*Figure 1B*) (*Gallardo et al., 2007*; *Lan et al., 2004*). It is possible that the low level of KDM1A remaining in some oocytes is due to delayed deletion of *Kdm1a*, though the reason for this potential delay is unknown.

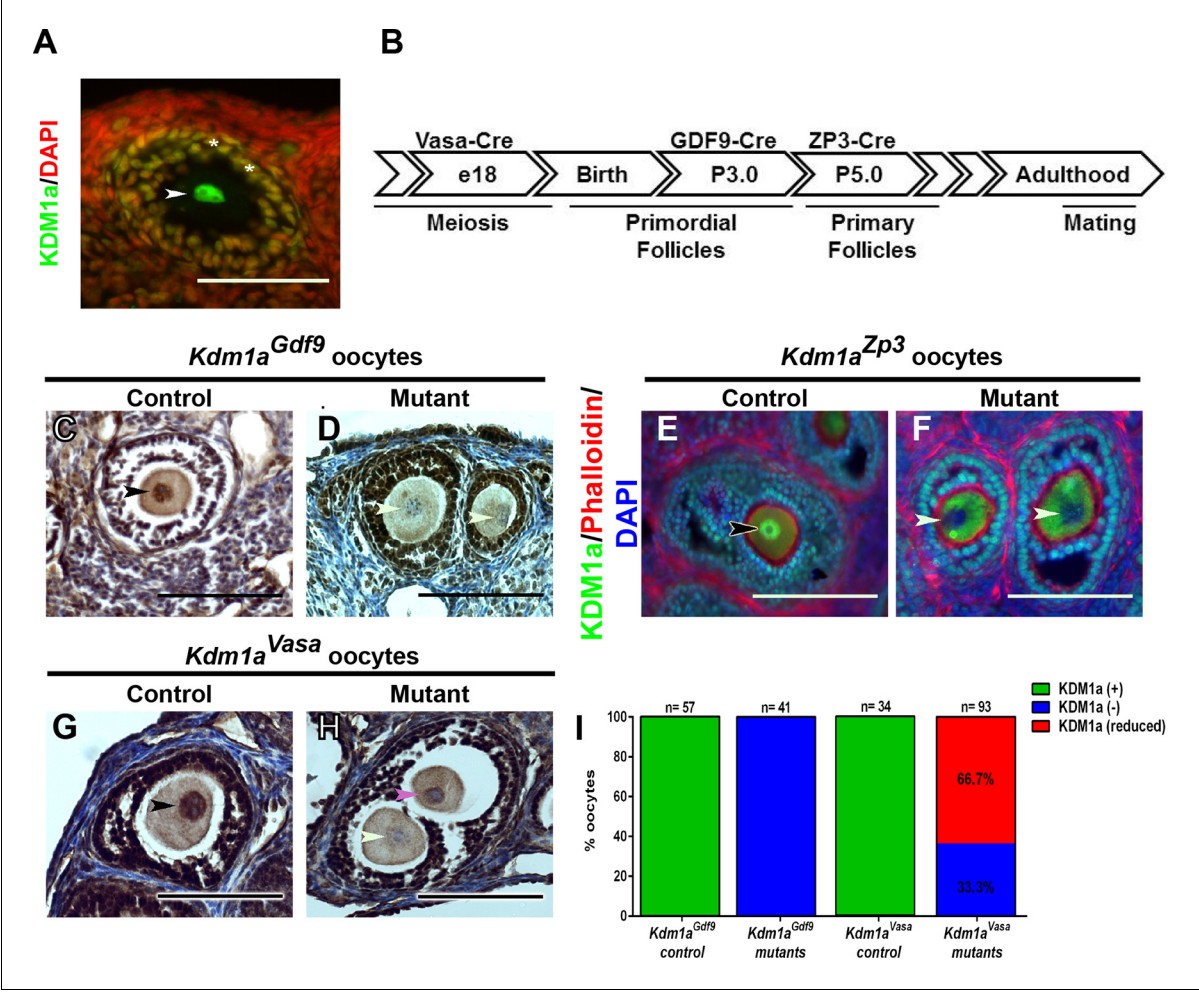

**Figure 1.** Maternal expression and coditional deletion of *Kdm1a* in mouse oocytes. (**A**) Wild-type mouse oocyte nucleus (white arrowhead) and surrounding follicle cells (white asterisks) stained with anti-KDM1A (green) antibody and DAPI (red). (**B**) Developmental timeline of maternal *Cre* expression (*Vasa-Cre, Gdf9-Cre* and *Zp3-Cre* transgenes) and corresponding oogenesis stages. (**C,D**) Immunohistochemistry (IHC) with anti-KDM1A (brown) antibody and hematoxylin (blue) showing KDM1A nuclear expression (black arrowhead) and absence of expression (white arrowheads) in *Kdm1a*^Gdf9^ control (**C**) and mutant (**D**) oocytes. (**E,F**) Immunofluorescence (IF) with anti-KDM1A (green) antibody, phalloidin (red) and DAPI (blue) showing KDM1A nuclear expression (black arrowhead) and absence of expression (white arrowheads) in *Kdm1a*^Zp3^ control (**E**) and mutant (**F**) oocytes. (**G,H**) IHC with anti-KDM1A (brown) antibody and hematoxylin (blue) showing KDM1A nuclear expression (black arrowhead), absence of expression (white arrowhead) and reduced expression (pink arrowhead) in *Kdm1a*^Vasa^ control (**G**) and mutant (**H**) oocytes. (**I**) Percentage of oocytes with KDM1A (green), reduced KDM1A (red) or no KDM1A blue) staining in *Kdm1a*^Gdf9^ and *Kdm1a*^Vasa^ heterozygous control versus mutant oocytes. Scale bars represent 50 µm. n=number of oocytes analyzed with percentages indicated for each category.

The following figure supplements are available for figure 1:

**Figure supplement 1.** KDM1A expression in stged oocytes.

**Figure supplement 2.** Generation of *Kdm1a* mutant and control animals.

## Loss of maternal KDM1A results in 1–2 cell embryonic arrest

To determine if there is a functional requirement for maternal KDM1A in mice, we crossed *Kdm1a-*^Vasa^, *Kdm1a*^Gdf9^ and *Kdm1a*^Zp3^ females to wild-type males to generate heterozygous offspring (*Figure 1—figure supplement 2*). In mice, zygotic transcription begins in the 1C embryo just prior to the first cleavage to the 2C stage (*Aoki et al., 1997*; *Hamatani et al., 2004*; *Xue et al., 2013*). The heterozygous offspring from the maternally deleted mothers have a normal *Kdm1a* gene on the paternal allele. Thus, crossing maternally deleted mothers to wild-type fathers enables us to isolate

the maternal function of KDM1A (Maternal-, Zygotic+, hereafter referred to as M-Z+). M-Z+ heterozygous embryos derived from $Kdm1a^{Gdf9}$ mutant mothers are hereafter referred to as $Kdm1a^{Gdf9}$ M-Z+ embryos, while M+Z+ heterozygous embryos derived from littermate control mothers that are *Cre* minus are hereafter referred to as $Kdm1a^{Gdf9}$ M+Z+ embryos. $Kdm1a^{Gdf9}$ M-Z+ embryos exhibit embryonic arrest at the 1-2C stage (*Figure 2—figure supplement 1A–I*). Specifically, in control $Kdm1a^{Gdf9}$ M+Z+ embryos at embryonic day 1.5 (e1.5), we observe 7% fragmented/degraded embryos, 65% 1-cell embryos and 28% 2-cell embryos (n=135, *Figure 2—figure supplement 1I*). In contrast, in $Kdm1a^{Gdf9}$ M-Z+ embryos at e1.5 we observe 40% fragmented/degraded embryos, 59% unfertilized oocytes or 1-cell embryos, and only 1% 2C embryos (n=134, *Figure 2—figure supplement 1I*). The vast majority of the non-degraded $Kdm1a^{Gdf9}$ M-Z+ embryos are clearly fertilized and arrested at the 1C stage. However, we do occasionally observe unfertilized oocytes. In addition, we sometimes observe embryos that are highly abnormal morphologically and are difficult to clearly assign to a particular category. As a result, we quantified these $Kdm1a^{Gdf9}$ M-Z+ embryos together. Nevertheless, compared to $Kdm1a^{Gdf9}$ M+Z+ embryos, $Kdm1a^{Gdf9}$ M-Z+ embryos have a large increase in the number of fragmented/degraded embryos at the expense of normal 2C embryos (*Figure 2—figure supplement 1I*). Also, the remaining 1C and 2C $Kdm1a^{Gdf9}$ M-Z+ embryos do not progress beyond the 1-2C stage, as we never observe any later stage embryos even at e2.5 (*Figure 2—figure supplement 1C,G,H*).

M-Z+ heterozygous embryos derived from $Kdm1a^{Zp3}$ mothers (hereafter referred to as $Kdm1a^{Zp3}$ M-Z+embryos) undergo a phenotype that is similar to $Kdm1a^{Gdf9}$ M-Z+ embryos. At e1.5, 95% (n=20) of control $Kdm1a^{Zp3}$ M+Z+ embryos have reached the 2C stage (*Figure 2E*). In contrast, at e1.5 only 35% (n=57) of $Kdm1a^{Zp3}$ M-Z+ embryos reach the 2C stage (*Figure 2E*), though this is still much higher than the 1% of $Kdm1a^{Gdf9}$ M-Z+ embryos that reach the 2C stage (*Figure 2—figure supplement 1I*). It is not clear why there is a difference in the number of $Kdm1a^{Zp3}$ versus $Kdm1a^{Gdf9}$ M-Z+ embryos that cleave to the 2C stage. However, this may be due to subtle differences in developmental timing, strain background, or *Cre* specific differences. In addition, we have observed two $Kdm1a^{Zp3}$ M-Z+ embryos that develop past the 2C stage to the 3C- and 4C- stage (*Figure 2—figure supplement 1J,K*), though like the $Kdm1a^{Gdf9}$ M-Z+ embryos, no $Kdm1a^{Zp3}$ M-Z+ embryos survive to the blastocyst stage (data not shown). Taken together, the results from $Kdm1a^{Gdf9}$ and $Kdm1a^{Zp3}$

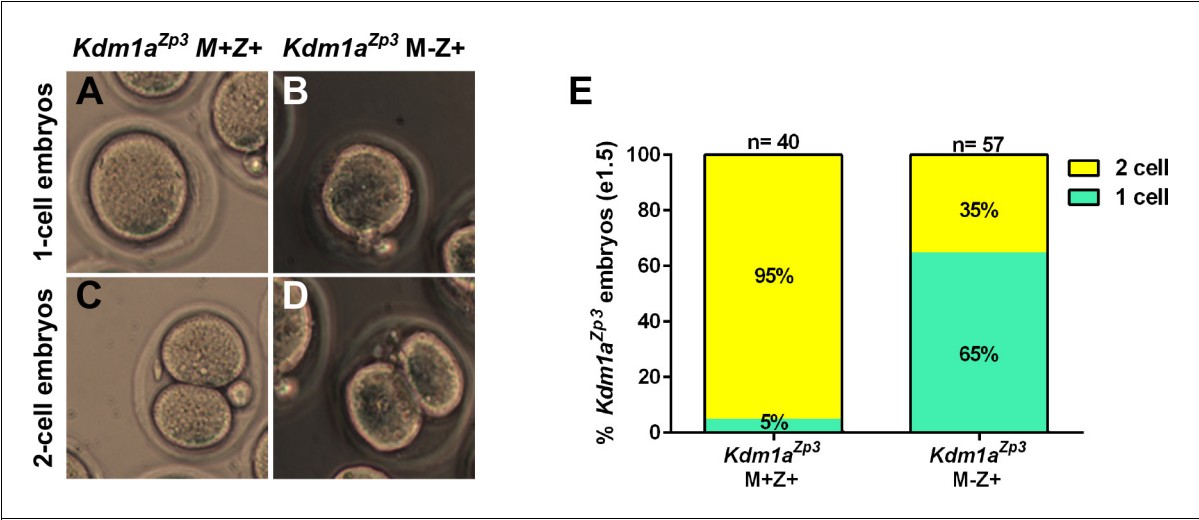

**Figure 2.** $Kdm1a^{Zp3}$ embryos arrest at the 1–2 cell stage. (A–D) Brightfield images of (A,C) M+Z+ and (B,D) M-Z+ 1- and 2-cell embryos derived from $Kdm1a^{Zp3}$ control and mutant mothers at e1.5. (E) Percentage of 1-cell (green) and 2-cell (yellow) embryos derived from $Kdm1a^{Zp3}$ control and mutant mothers at e1.5. n = 40 for $Kdm1a^{Zp3}$ M+Z+ embryos from 3 litters. n = 57 for $Kdm1a^{Zp3}$ M-Z+ embryos from 6 litters.

The following figure supplement is available for figure 2:

**Figure supplement 1.** Lack of normal $Kdm1a^{Gdf9}$ and $Kdm1a^{Zp3}$ embryos at embryonic day 1.5 and 2.5.

mice suggest that the embryonic arrest, primarily at the 1-2C stages is the KDM1A maternal loss-of-function phenotype.

## Loss of maternal KDM1A results in a failure to undergo the MZT

To determine the maternal function of KDM1A, we considered two possibilities; (1) that KDM1A affects transcription in the oocyte, or (2) that maternal KDM1A affects transcription post-fertilization during the MZT. To determine if KDM1A affects transcription in the oocyte, we first compared the transcriptome of control *Kdm1a^fl/fl^* (*Macfarlan et al., 2012*) versus *Kdm1a^Zp3^* mutant oocytes. We found relatively few transcriptional changes, with 195 genes over expressed and 281 genes under expressed in *Kdm1a^Zp3^* oocytes (*Figure 3A*, *Figure 3—source data 1A*, *Figure 3—figure supplement 5*). In addition, amongst the 875 repeat types extracted from the RepeatMasker database (including LINEs, SINEs, etc.) there were only 2 repeat families that were activated and 2 repressed in *Kdm1a^Zp3^* oocytes (*Figure 3A*, *Figure 3—source data 1A*, *Figure 3—figure supplement 5*). Manual inspection revealed that few of these genes/repeats had both large fold-changes and low p-values, with the exception of the Mt1 gene, which was very highly activated in *Kdm1a^Zp3^* oocytes. Furthermore, we found several mitochondrially encoded genes (like mtCo3 and Atpase6) that had small but significantly reduced levels in *Kdm1a^Zp3^* oocytes. These mitochondrial genes are unlikely to be directly regulated by KDM1A, but their reduced levels could indicate a subtle metabolic defect in the mutants. Taken together, these results demonstrate that loss of KDM1A has little effect on transcription in oocytes.

Since we did not see a significant affect on oocytes, we considered the possibility that loss of KDM1A affects transcription post-fertilization during the MZT. To determine if this is the case, we isolated fertilized *Kdm1a^Zp3^* M-Z+ embryos that cleaved to the 2C stage, and compared them to *Kdm1a^fl/fl^* M+Z+ 2C embryos (*Macfarlan et al., 2012*). We chose to focus on embryos from *Kdm1a^Zp3^* mothers because a substantially higher proportion of these embryos reach the 2C stage. We found a dramatic alteration of the transcriptome, with 1527 genes over expressed and 2794 genes under expressed in *Kdm1a^Zp3^* M-Z+ 2C embryos (*Figure 3B*, *Figure 3—source data 1B*). In addition, there were 103 repeat families that were repressed in *Kdm1a^Zp3^* 2C M-Z+ embryos, compared with only 5 that were activated (*Figure 3B*, *Figure 3—source data 1B*). These findings are in contrast to what has been observed in *Kdm1a* mutant ES cells that display a general de-repression of both genes and repeats (*Macfarlan et al., 2011*; *Macfarlan et al., 2012*). GO analysis of over and under expressed genes in *Kdm1a^Zp3^* 2C M-Z+ embryos demonstrated an enrichment of genes normally expressed in unfertilized oocytes, and a reduction in genes associated with embryonic development and tissue specific gene expression. This suggests that *Kdm1a^Zp3^* M-Z 2C embryos may fail to undergo the MZT (*Figure 3B*, *Figure 3—source data 1B*).

To test this hypothesis further, we compared the expression of mRNAs from control *Kdm1a^fl/fl^* M+Z + 2C embryos (*Macfarlan et al., 2012*) or *Kdm1a^Zp3^* M-Z+ 2C embryos with *Kdm1a^fl/fl^* oocytes (*Macfarlan et al., 2012*). If the *Kdm1a^Zp3^* M-Z+ 2C embryos fail to undergo the MZT, then we would expect them to be more similar to *Kdm1a^fl/fl^* oocytes than *Kdm1a^fl/fl^* M+Z+ 2C embryos. In *Kdm1a^fl/fl^* M+Z+ 2C embryos relative to *Kdm1a^fl/fl^* oocytes, there are massive changes in the mRNA profile, with >3,000 genes (and 387 repeat families) becoming zygotically activated and ~3000 maternal genes (and 17 repeat families) which are suppressed. This demonstrates that MZT has occurred (*Figure 3C*, *Figure 3—source data 1C*). In contrast, *Kdm1a^Zp3^* M-Z+ 2C embryos fail to properly activate the zygotic genome, with only 666 genes becoming activated (and 110 repeat families), and only ~1200 maternal genes repressed (13 repeat families) (*Figure 3D*, *Figure 3—source data 1D*). Hierarchical clustering of *Kdm1a^fl/fl^* oocytes, *Kdm1a^Zp3^* oocytes, *Kdm1a^fl/fl^* M+Z+ 2C embryos and *Kdm1a^Zp3^* M-Z+ 2C embryos confirms this failure to activate the zygotic genome. Specifically, though *Kdm1a^fl/fl^* oocytes are similar to *Kdm1a^Zp3^* oocytes, *Kdm1a^Zp3^* M-Z+ 2C embryos are more similar to *Kdm1a^fl/fl^* oocytes than *Kdm1a^fl/fl^* M+Z+ 2C embryos (*Figure 3E*). In addition, principal component analysis (PCA) demonstrated that ~40% of the variance between *Kdm1a^fl/fl^* oocytes, *Kdm1a^fl/fl^* M+Z+ 2C embryos, and *Kdm1a^Zp3^* M-Z+ 2C embryos can be explained by a single component (PC1, *Figure 3—figure supplement 2*), and the heat maps generated from PC1 genes confirm that *Kdm1a^Zp3^* M-Z+ 2C embryos have expression profiles more similar to *Kdm1a^fl/fl^* oocytes than *Kdm1a^fl/fl^* M+Z+ 2C embryos (*Figure 3F*). GO analysis demonstrated that PC1 genes fall into two categories, those associated with unfertilized oocytes and are highly expressed in oocytes and *Kdm1a^Zp3^* M-Z+ 2C embryos, and those associated with early embryonic development which are enriched only in *Kdm1a^fl/fl^* M+Z+ 2C embryos

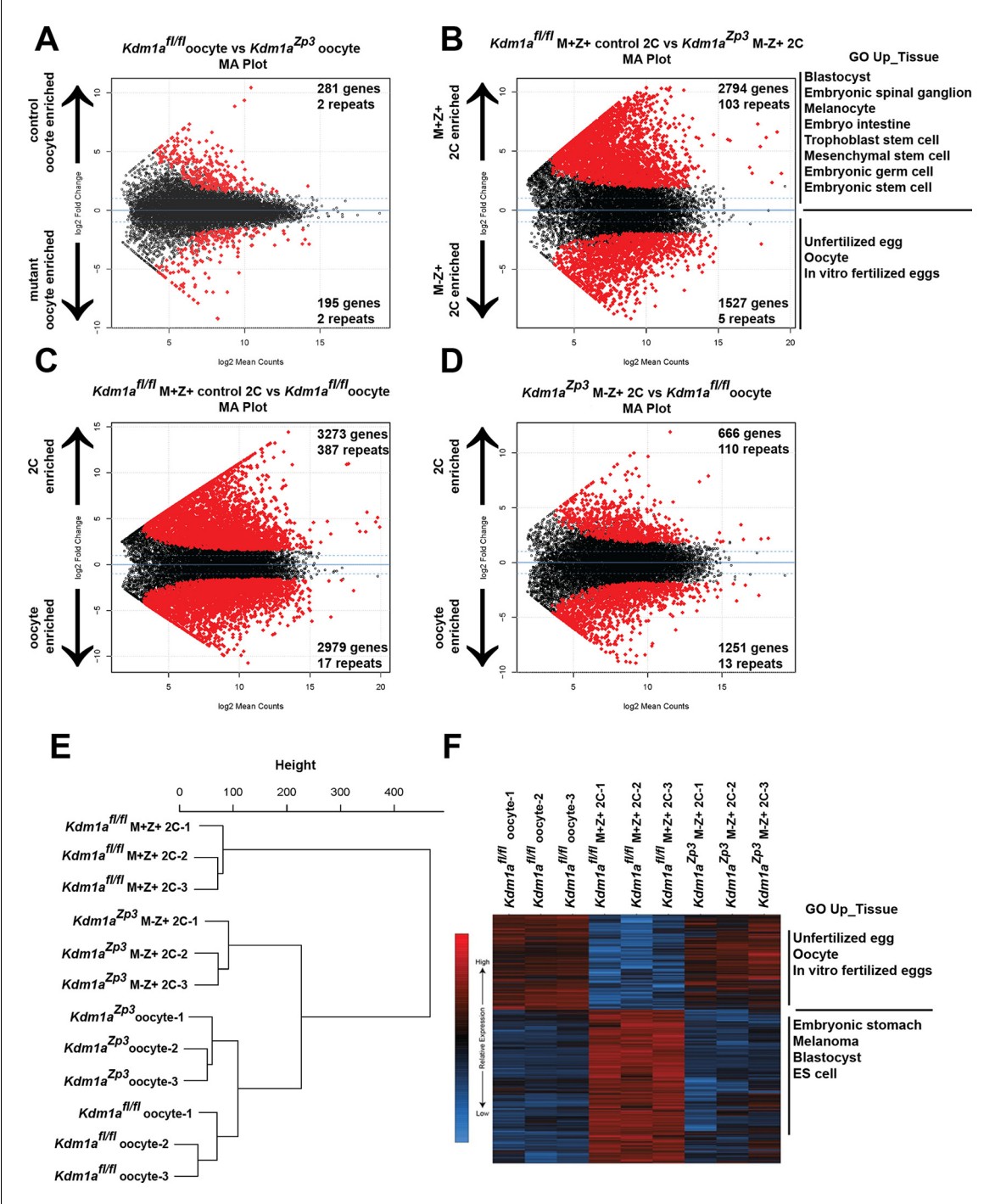

**Figure 3.** The MZT is impaired in *Kdm1a^Zp3* mutants. (**A,B**) Differential expression of mRNAs in *Kdm1a^fl/fl* versus *Kdm1a^Zp3* oocytes (**A**) or *Kdm1a^fl/fl* M+Z+ versus *Kdm1a^Zp3* M-Z+ 2C embryos (**B**) as determined by RNA-seq. Genes/repeats highlighted in red are significant with the number of significant gene/repeats show. GO enrichment using the Up_tissue database was performed on *Kdm1a^fl/fl* M+Z+ 2C enriched and *Kdm1a^Zp3* M-Z+ 2C enriched mRNAs, with a list of the most enriched categories displayed. (**C,D**) Differential expression of mRNAs in *Kdm1a^fl/fl* M+Z+ 2C embryos versus *Kdm1a^fl/fl* oocytes (**C**) or *Kdm1a^Zp3* M-Z+ 2C embryos versus *Kdm1a^fl/fl* oocytes (**D**). The numbers of zygotically activated (2C enriched) genes/repeats and zygotically repressed (oocyte enriched) genes/repeats are highlighted in each comparison. (**E**) Hierarchical cluster dendrogram of transcriptomes in *Kdm1a^fl/fl* oocytes, *Kdm1a^Zp3* oocytes, *Kdm1a^fl/fl* M+Z+ 2C embryos, and *Kdm1a^Zp3* M-Z+ 2C embryos. (**F**) Heat map of gene expression of principal component 1 (PC1) genes in *Kdm1a^fl/fl* oocytes, *Kdm1a^fl/fl* M+Z+ 2C embryos, and *Kdm1a^Zp3* M-Z+ 2C embryos. The most GO Up_tissue enriched terms are displayed for the 2 categories of PC1 genes.

*Figure 3 continued*

The following source data and figure supplements are available for figure 3:

**Source data 1.** Gene Table comparison of $Kdm1a^{fl/fl}$ oocytes and $Kdm1a^{Zp3}$ oocytes.
**Source data 2.** Gene Table comparison of $Kdm1a^{Zp3}$ M+Z+ 2C and $Kdm1a^{Zp3}$ M-Z+ 2C embryos.
**Source data 3.** Gene Table comparison of $Kdm1a^{Zp3}$ M+Z+ 2C and $Kdm1a^{fl/fl}$ oocytes.
**Source data 4.** Gene Table comparison of $Kdm1a^{Zp3}$ M-Z+ 2C and $Kdm1a^{fl/fl}$ oocytes.
**Figure supplement 1.** The MZT is impaired in $Kdm1a^{Zp3}$ mutants.
**Figure supplement 2.** Principal component analysis of $Kdm1a^{Zp3}$ 2C embryos.
**Figure supplement 3.** Expression of epigenetic regulators in $Kdm1a^{Zp3}$ 2C embryos.
**Figure supplement 4.** Relative expression of epigenetic regulators in $Kdm1a^{Zp3}$ 2C embryos.
**Figure supplement 5.** Expression of epigenetic regulators in $Kdm1a^{fl/fl}$ and $Kdm1a^{Zp3}$ oocytes.

(*Figure 3F*). Taken together, these data confirm that the major defect in the transcriptome of $Kdm1a^{Zp3}$ M-Z+ 2C embryos is a failure to undergo the MZT.

## Deletion of *Kdm1a* maternally with *Vasa-Cre* results in a hypomorphic phenotype

Deletion of *Kdm1a* with either *Gdf9-Cre* or *Zp3-Cre* results in 100% of oocytes completely lacking KDM1A (*Figure 1C–F*). In contrast, 66.7% of $Kdm1a^{Vasa}$ mutant oocytes retain some KDM1A protein, though the amount is much lower than heterozygous control oocytes (*Figure 1G–I*). To determine if this lower amount of KDM1A retained in some $Kdm1a^{Vasa}$ mutant oocytes gives rise to a hypomorphic phenotype, we mated $Kdm1a^{Vasa}$ females to wild-type males to generate heterozygous M-Z+ offspring (*Figure 1—figure supplement 2*). As is the case with $Kdm1a^{Gdf9}$ and $Kdm1a^{Zp3}$ M-Z+ offspring, these heterozygous M-Z+ offspring (hereafter referred to as $Kdm1a^{Vasa}$ M-Z+ embryos) have a normal paternal *Kdm1a* allele. Thus, any phenotypic effects in these $Kdm1a^{Vasa}$ heterozygous M-Z+ offspring are due to reduced maternal KDM1A.

Similar to $Kdm1a^{Gdf9}$ and $Kdm1a^{Zp3}$ M-Z+ embryos, the majority of $Kdm1a^{Vasa}$ M-Z+ embryos arrest prior to the blastocyst stage (*Figure 4A–E*). In control $Kdm1a^{Vasa}$ M+Z+ embryos at e3.5, 55% of embryos have reached the blastocyst stage, 35% are multicellular (>2C) and 10% are fragmented/degraded (n=37, *Figure 4E*). In contrast, at e3.5 85% of the $Kdm1a^{Vasa}$ M-Z+ embryos have fragmented/degraded (n=35, *Figure 4E*). However, unlike $Kdm1a^{Gdf9}$ and $Kdm1a^{Zp3}$ M-Z+ embryos, which undergo complete embryonic arrest at the 1-2C stage, by e3.5 6% of $Kdm1a^{Vasa}$ M-Z+ embryos have reached the multicellular stage (>2C) and 5% are blastocysts (n=35, *Figure 4E*). Remarkably, some of these $Kdm1a^{Vasa}$ M-Z+ embryos survive to birth. The average litter size born from $Kdm1a^{Vasa}$ mutant mothers is 2.3 (n=20), versus 6.2 (n=32) born from littermate control mothers (*Figure 4F*). However, the vast majority of time vaginal plugged $Kdm1a^{Vasa}$ mutant mothers give rise to no viable progeny. Thus, this average litter size is undoubtedly a vast overestimate of the survival of $Kdm1a^{Vasa}$ M-Z+ embryos overall.

The transition from in utero development to postnatal development, during the first 48 hr immediately following birth, is a highly stressful time for mice. If a mouse has a subtle defect during embryogenesis, the newborn pup may die perinatally. This is reflected in the large number of mouse models that exhibit perinatal lethality (*Turgeon and Meloche, 2009*). Therefore, to determine if lower maternal KDM1A might give rise to more subtle embryonic defects, we carefully monitored the first 48 hr following birth in $Kdm1a^{Vasa}$ M-Z+ progeny. Consistent with this being a major developmental transition, even in genotypically normal mice, we observe that 5% of offspring (n=216) born from littermate $Kdm1a^{Vasa}$ control mothers die perinatally during the first 48 hr after

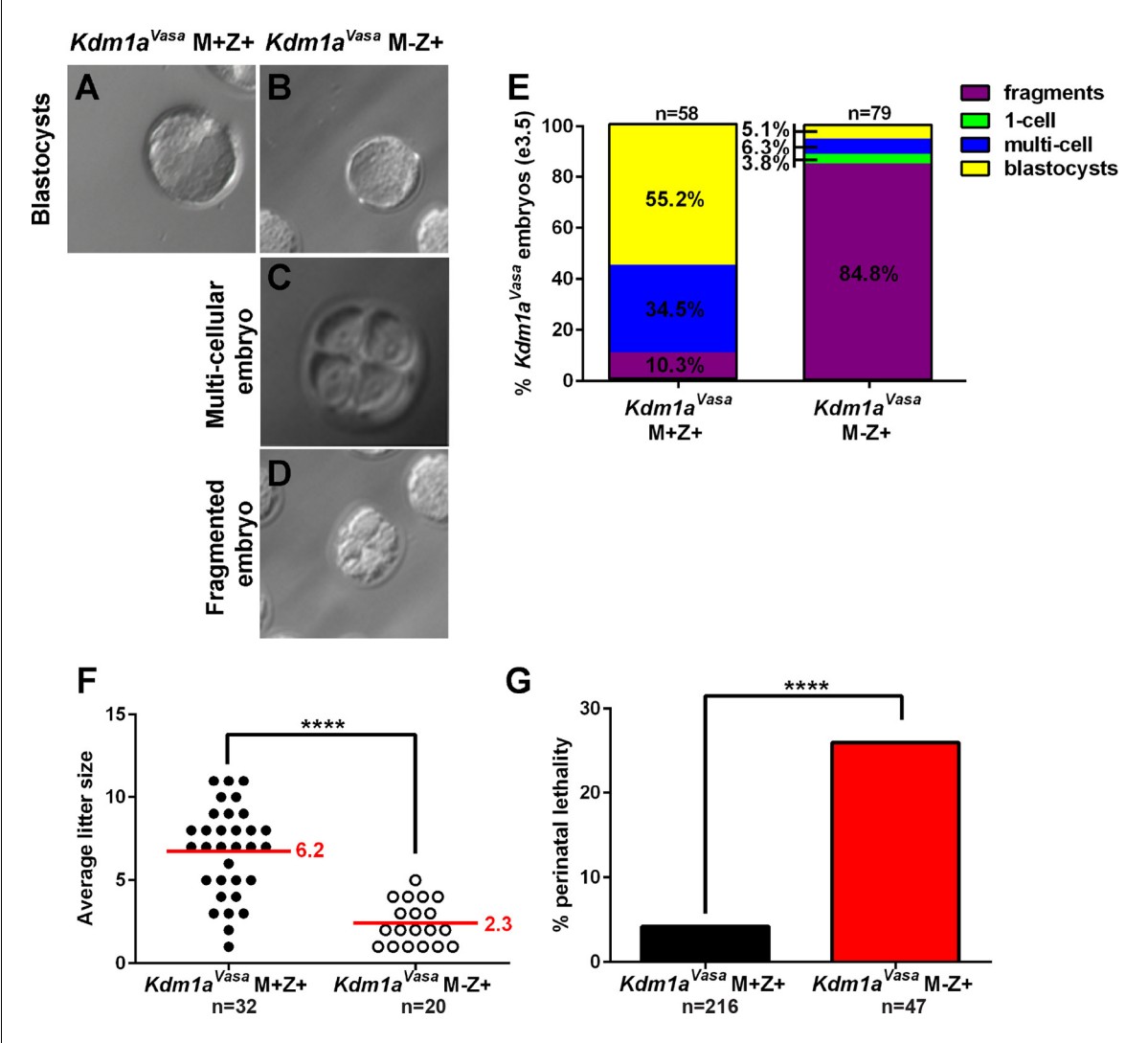

**Figure 4.** Hypomorphic phenotype in *Kdm1a*[Vasa] progeny. (**A–D**) Brightfield images of M+Z+. (**A**) and M-Z+ (**B–D**) embryos derived from *Kdm1a*[Vasa] control and mutant mothers at embryonic day 3.5 (e3.5). Panels show blastocysts (**A,B**), a multicellular embryo (**C**) and a fragmented embryo (**D**). (**E**) Percentage of fragmented (purple), 1-cell (green), multi-cellular (blue) and blastocyst (yellow) embryos from *Kdm1a*[Vasa] control and mutant mothers at e3.5. n = 58 for *Kdm1a*[Vasa] M+Z+ embryos from 7 litters. n = 79 for *Kdm1a*[Vasa] M-Z+ embryos from 10 litters. (**F**) Litter sizes of *Kdm1a*[Vasa] control and mutant mothers. Average litter size for each indicated by red line. Each circle indicates one litter and n=number of litters analyzed. p-values calculated using an unpaired t-test with **** = p<0.0001 indicating statistical significance. (**G**) Percentage of newborn pups from *Kdm1a*[Vasa] heterozygous control and mutant mothers that died perinatally. n = number of litters analyzed. p-values calculated using an unpaired t-test with **** = p<0.0001 indicating statistical significance.

birth (*Figure 4G*). However, in *Kdm1a*[Vasa] M-Z+ progeny, the percentage of perinatal lethality is significantly increased (26%, n=47) (*Figure 4G*). It is unclear why these animals are dying perinatally. However, this increase in perinatal lethality is consistent with these animals having more subtle developmental defects.

## *Kdm1a*[Vasa] M-Z+ progeny exhibit abnormal behavior

Of the small number of M-Z+ animals that are born from *Kdm1a*[Vasa] mutant mothers, nearly 1/3 die perinatally in the first 48 hr after birth. However, the remaining 2/3 survive to adulthood and appear to be morphologically normal. Notably, because of the embryonic arrest and the perinatal lethality, this remaining 2/3 is a very small number of animals (n=10 from 8 crosses). Since progeny from *Kdm1a*[Vasa] mutant mothers have defects that can lead to perinatal lethality, we considered the

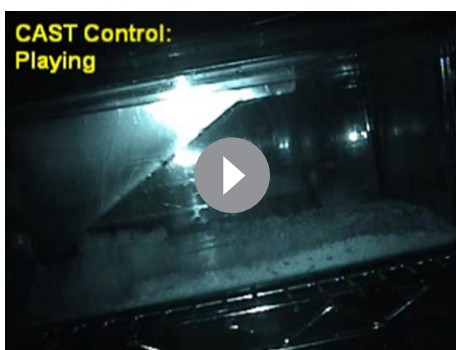

**Video 1.** Stereotypical and abnormal behaviors in *Kdm1a*<sup>Vasa</sup> M-Z+ adults. 0:00-0:27 *M. castaneus* (CAST) control playing behavior. 0:27-0:40 *M. castaneus* (CAST) control eating. 0:40-1:03 (right) *Kdm1a*<sup>Vasa</sup> M-Z+ adult food harassing behavior. 1:03-1:28 (right) *Kdm1a*<sup>Vasa</sup> M-Z+ adult abnormal scratching behavior. 1:28-1:45 (right) *Kdm1a*<sup>Vasa</sup> M-Z+ adult food harassing behavior and (left) abnormal digging behavior.

possibility that the surviving animals from these *Kdm1a*<sup>Vasa</sup> mutant mothers (hereafter referred to as *Kdm1a*<sup>Vasa</sup> M-Z+ adults) could have phenotypic defects as well. To analyze these surviving animals, we crossed *Kdm1a*<sup>Vasa</sup> mutant mothers (C57BL/6 background) to *M. castaneus* (CAST) fathers. This enabled us to use polymorphisms to examine subsequent molecular defects in a parent of origin specific fashion. Importantly, F1 hybrid (B6/CAST) offspring from *Kdm1a*<sup>Vasa</sup> mutant mothers have the same hypomorphic phenotype that we observe in a B6 background, where a small number of animals survive until birth (average litter size=1.6, n=6 litters from 8 crosses).

Mating the surviving *Kdm1a*<sup>Vasa</sup> M-Z+ adults produced progeny, suggesting no gross defects in the specification of the germline or mating behavior. However, when we observed these *Kdm1a*<sup>Vasa</sup> M-Z+ adults closely, we noticed that they exhibit abnormal behaviors (*Figure 5—figure supplement 1A*, *Video 1*). These abnormal behaviors include excessive scratching and digging, along with food harassing, where animals grind up all of the remaining food in the food hopper and incorporate it into the bedding (*Figure 5A–C*). To quantify the food harassing behavior, we monitored both the weight of the food in the food hopper and the height of the bedding in the cage. Over a 3-day period, each individual *Kdm1a*<sup>Vasa</sup> M-Z+ adult almost completely depleted the food in the food hopper due to food grinding and harassing (*Figure 5D* and *Figure 5—figure supplement 1B–D*). This resulted in a correspondingly large increase in the height of the bedding over a 6-day period (*Figure 5E*). These effects are not observed in *Kdm1a*<sup>Vasa</sup> mutant mothers or the *M. castaneus* animals to which they were mated (*Figure 5D,E* and *Figure 5—figure supplement 1B,E*), indicating that the behavior may be due to a maternal effect. To determine if this behavior is due to a maternal effect, we intercrossed the affected *Kdm1a*<sup>Vasa</sup> M-Z+ adults. If the food-grinding behavior is due to a maternal effect, then mating two affected *Kdm1a*<sup>Vasa</sup> M-Z+ adults should not produce progeny with abnormal behavior. This is because the *Kdm1a*<sup>Vasa</sup> M-Z+ adults used as mothers in these crosses have a normal *Kdm1a* allele (-/+) so the resulting intercrossed progeny (hereafter referred to as F2 intercrossed M+Z+ adults) are all M+. Intercrossing affected M-Z+ F1 hybrids produced no F2 intercrossed M+Z+ adults with the food-grinding behavior, suggesting that the food-grinding behavior is dependent upon a maternal effect (*Figure 5D,E* and *Figure 5—figure supplement 1D,G*). Nevertheless, it remains possible that the F1 hybrid background also contributes to the abnormal behavior. To test this possibility, we crossed control littermates of *Kdm1a*<sup>Vasa</sup> mothers to CAST fathers to generate control B6/CAST hybrids (hereafter referred to as B6/CAST M+Z+ controls). These control hybrids exhibit modest food-grinding behavior, suggesting that the hybrid strain background likely also contributes to the maternal food-grinding defect (*Figure 5D,E* and *Figure 5—figure supplement 1C,F*). However, *Kdm1a*<sup>Vasa</sup> M-Z+ adults are significantly more affected than B6/CAST M+Z+ hybrid controls (*Figure 5D,E* and *Figure 5—figure supplement 1C,F*), suggesting that the abnormal food-grinding behavior is predominantly due to a maternal effect. Importantly, we also do not observe any *Kdm1a* haploinsufficiency defects. For example, the F2 intercrossed M+Z+ adults that are heterozygous for *Kdm1a* do not exhibit this abnormal behavior (*Figure 5D,E*; *Figure 5—figure supplement 1D,G*). This suggests that the abnormal behavior is not due to *Kdm1a* haploinsufficiency. Finally, behavioral defects were observed in *Kdm1a*<sup>Vasa</sup> M-Z+ survivors obtained from litters ranging from one to four animals. This suggests that the maternal behavioral abnormalities are not simply due to reduced litter size.

Observations of *Kdm1a*<sup>Vasa</sup> M-Z+ adults indicated that these mice also exhibit excessive digging (*Figure 5—figure supplement 1A*, *Video 1*). To quantify this behavior, we assayed *Kdm1a*<sup>Vasa</sup> M-Z+ adults and controls in a marble-burying assay. In this assay, individual mice are placed in a cage in

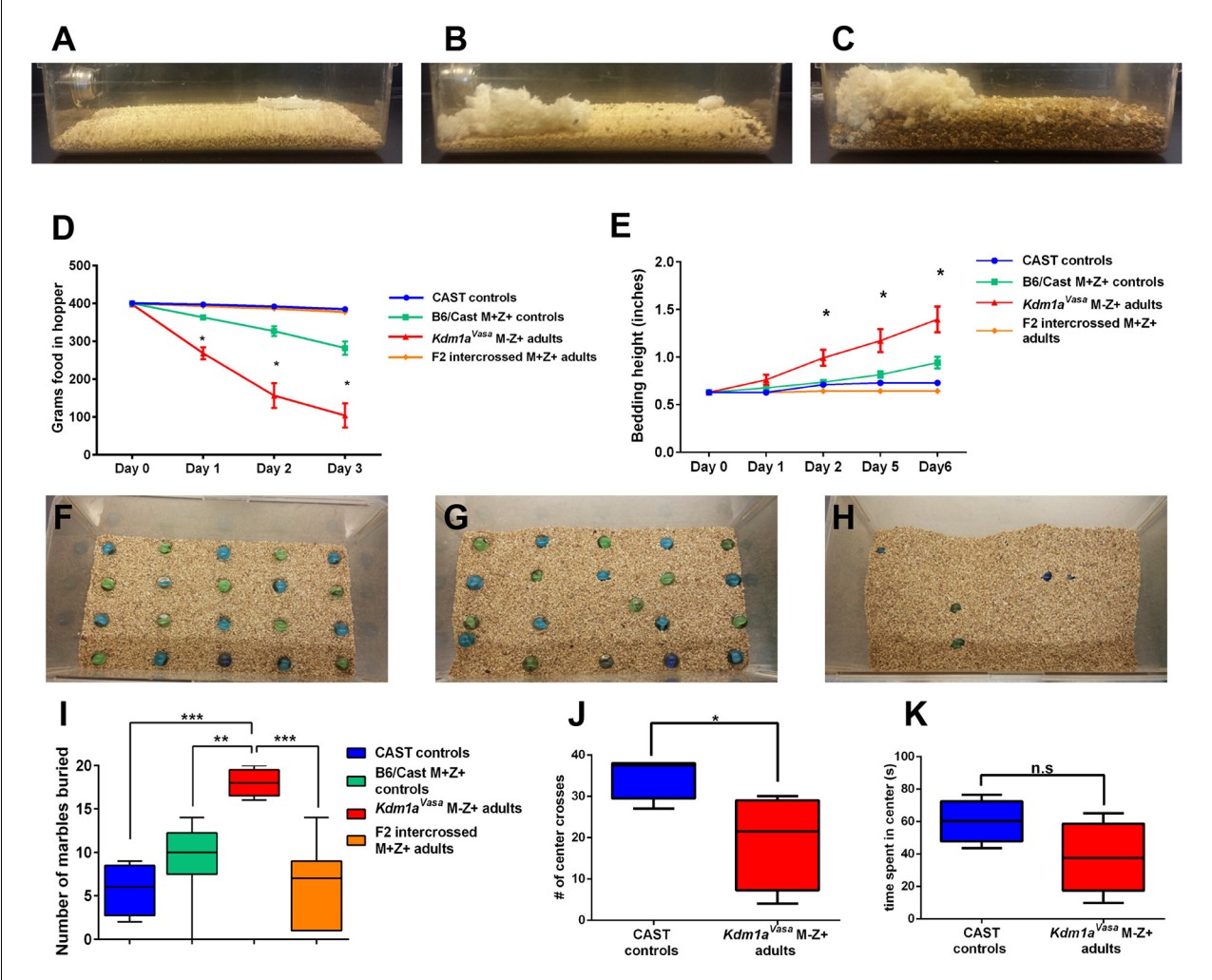

**Figure 5.** Abnormal behaviors in *Kdm1a^Vasa^* M-Z+ adults. (A–C) Mouse cages at day 0 (A) and day 8 (B) from *M. castaneus* (CAST) controls compared to day 6 (C) from a *Kdm1a^Vasa^* M-Z+ adult. (D) Quantification of change in weight of food in the hopper from CAST controls, B6/CAST hybrid M+Z+ controls, and F2 intercrossed M+Z+ adults versus *Kdm1a^Vasa^* M-Z+ adults. Data are shown as mean for each day with error bars indicating ± S.E.M. (E) Quantification of change in bedding height from CAST controls, B6/CAST hybrid M+Z+ controls, and F2 intercrossed M+Z+ adults versus *Kdm1a^Vasa^* M-Z+ adults. Data are shown as mean for each day with error bars indicating ± S.E.M. (F–H) Mouse cages before (F) and after (G,H) the marble burying assay was performed on a CAST control (G) compared to a *Kdm1a^Vasa^* M-Z+ adult (H). (I) Quantification of the number of marbles buried during the marble burying assay performed on CAST controls, B6/CAST hybrid M+Z+ controls, and F2 intercrossed M+Z+ adults versus *Kdm1a^Vasa^* M-Z+ adults. Data are shown as quartiles with error bars indicating ± S.E.M. (J,K) Open field test performance in CAST controls versus *Kdm1a^Vasa^* M-Z+ adults scored by number of center crosses (J) and time spent in center of cage (K). Data are shown as quartiles with error bars indicating ± S.E.M. p-values calculated using an unpaired t-test with n.s. indicating $p > 0.05$, * = $p < 0.05$, ** = $p < 0.005$, *** = $p < 0.0005$. All asterisks indicate statistical significance.

The following figure supplement is available for figure 5:

**Figure supplement 1.** Abnormal behaviors in individual *Kdm1a^Vasa^* M-Z+ adults.

which 20 marbles are arrayed on top of the bedding. Mice exhibiting excessive digging behavior will bury higher numbers of marbles. For example, *Slitrk5* obsessive-compulsive mice bury approximately 50% of the marbles in 30 min, compared to approximately 25% in controls (*Shmelkov et al., 2010*).

*Kdm1a^Vasa^* M-Z+ adults exhibit striking behavior in the marble-burying assay. Compared to *M. castaneus* controls, which bury 25% in 25 min (*Figure 5G,I*, *Video 2*), *Kdm1a^Vasa^* M-Z+ adults bury 90% in the same time period (*Figure 5H,I*, *Video 2*). Similar to what was observed in the food-grinding assays, we do not observe this behavior in F2 intercrossed M+Z+ adults (average 7 marbles,

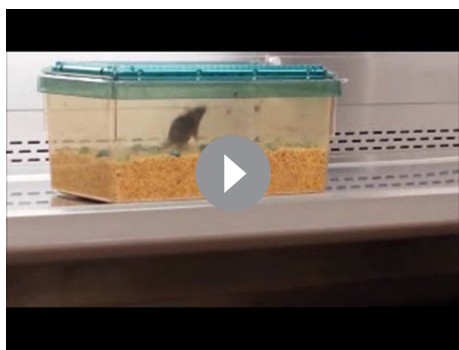

**Video 2.** Marble burying in *Kdm1a^Vasa^* adults. 0:00-0:23 Clip 1 of *M. castaneus* (CAST) control in marble burying assay. 0:23-0:44 Clip 2 of *M. castaneus* (CAST) control in marble burying assay. 0:44-1:22 Clip 1 of *Kdm1a^Vasa^* M-Z+ adult in marble burying assay. 1:22-1:36 Clip 2 of *Kdm1a^Vasa^* M-Z+ adult in marble burying assay. 1:36-2:13 Clip 3 of *Kdm1a^Vasa^* M-Z+ adult in marble burying assay.

## imprinting defects

The perinatal lethality and behavioral defects observed in *Kdm1a^Vasa^* M-Z+ progeny are remarkable because these animals have a normal *Kdm1a* allele. Thus the defects that we observe in *Kdm1a^Vasa^* M-Z+ progeny must be due to a heritable effect originating from the low level of maternal KDM1A. As a result, we sought to determine the nature of this heritable defect. The *Kdm1a* homolog *Kdm1b* is expressed maternally in mice, and loss of *Kdm1b* results in a heritable embryonic lethality defect associated with a failure to maternally acquire DNA methylation at imprinted genes (*Ciccone et al., 2009*; *Stewart et al., 2015*). Therefore, we considered the possibility that the heritable defects could be due to DNA methylation defects at these loci, or in maternally methylated imprinted loci unaffected by the loss of KDM1B. In addition, in our RNA-seq data, we observe the misregulation of multiple genes that could potentially affect DNA methylation at imprinted loci. For example, four genes that are known to maternally affect DNA methylation at imprinted loci, *Tet1, Trim28, Zfp57* and *Dppa3/Stella* (*Dawlaty et al., 2013*; *Li et al., 2008*; *Messerschmidt et al., 2012*; *Nakamura et al., 2007*; *Yamaguchi et al., 2013*), are all misregulated in *Kdm1a^Zp3^* M-Z+mutants (*Tet1:* -3.8 fold, *Trim28:* -3.3 fold, *Zfp57:* +3.2 fold and *Dppa3/Stella:* +.58 fold, *Figure 3—figure supplement 3*; *Figure 3—source data 1B*, quantitative RT-PCR validation of *Trim28, Zfp57* and

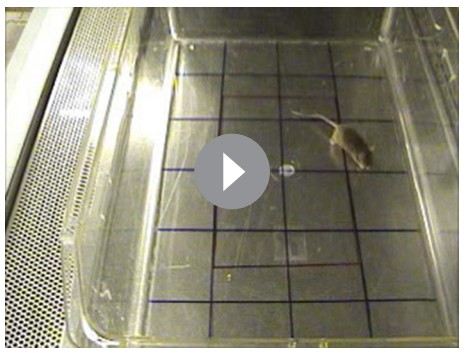

**Video 3.** Open field test on *Kdm1a^Vasa^* M-Z+ adults. 0:00-0:30 *M. castaneus* control in open field test. 0:30-0:56 *Kdm1a^Vasa^* M-Z+ adult in open field test.

*Figure 5I*), but we do see slightly elevated marble-burying in B6/CAST hybrid controls (average 10, *Figure 5I*). Nevertheless, as in the food-grinding assay, *Kdm1a^Vasa^* M-Z+ adults are significantly more affected than B6/CAST hybrid M+Z+ controls (*Figure 5D,E* and *Figure 5—figure supplement 1C,F*). Taken together, these data indicate that the excessive digging behavior is also predominantly due to a maternal effect.

Finally, we also performed the open-field test on *Kdm1a^Vasa^* M-Z+ adults. In this assay, animals are placed in a much larger space than they are accustomed to. Animals that are more anxious or fearful cross the middle of cage fewer times and spend less time overall in the center (*Video 3*). Consistent with the behavioral defects that we observe in food-grinding and marble-burying, *Kdm1a^Vasa^* M-Z+ adults perform significantly fewer movements across the middle of the cage (*Figure 5J*) and spend less time in the center overall than CAST controls (*Figure 5K*).

## *Kdm1a^Vasa^* M-Z+ progeny have

*Dppa3/Stella* in *Figure 3—figure supplement 4*). Additionally, the DNA methyltransferase DNMT1 is overexpressed in *Kdm1a^Zp3^* M-Z+ mutants (*Dnmt1:* +2.1 fold, *Figure 3—figure supplement 3*; *Figure 3—source data 1B*). DNMT1 is thought to primarily act as the maintenance DNA methyltransferase (*Li et al., 1992*; *Pradhan et al., 1999*), but when overexpressed, DNMT1 has been shown to have *de novo* DNA methyltransferase activity (*Vertino et al., 1996*). Taken together, these observations raise the possibility that DNA methylation and monoallelic expression at imprinted genes could be altered in *Kdm1a^Vasa^* M-Z+ progeny, either directly or indirectly. Interestingly, despite the clear loss of KDM1A in *Kdm1a^Zp3^* mutant oocytes, the regulatory proteins that are affected by the loss of

KDM1A are either not expressed in oocytes (Tet1) or not affected until after fertilization (*Trim28, Zfp57, Dppa3/Stella* and *Dnmt1, Figure 3—figure supplements 4,5*). This is consistent with our previous observations that KDM1A primarily affects the MZT.

To investigate DNA methylation and the expression of imprinted genes in *Kdm1a^Vasa^* M-Z+ offspring, we performed bisulfite analysis and qRT-PCR on perinatal lethal *Kdm1a^Vasa^* M-Z+ pups. We analyzed imprinted that are both maternally (*Zac1, Impact, Igf2r, Mest, Snrpn*) and paternally methylated (*H19*). For these analyses, we took advantage of B6/CAST polymorphisms to determine whether individual alleles in F1 hybrid offspring came from the mother or the father. These F1 hybrid pups, derived from hypomorphic B6 *Kdm1a^Vasa^* mothers mated to wild-type CAST fathers, are denoted as maternal effect progeny 1 and 2 (MEP1 and MEP2), and were compared to two stage matched B6/CAST hybrid controls.

Two imprinted loci, *Zac1* and *Impact*, are severely affected in *Kdm1a^Vasa^* offspring. At *Zac1,* DNA methylation is inappropriately acquired on the paternal allele in both MEP, though to a slightly lesser extent in MEP2 (*Figure 6A*). This is associated with significantly less expression of *Zac1* in both MEP (*Figure 6B*). Moreover, *Zac1* is normally expressed exclusively from the paternal allele (*Figure 6C*). However, in MEP1 *Zac1* is now expressed exclusively from the maternal allele, while in MEP2, the slightly lower level of inappropriate paternal methylation is associated with biallelic expression (*Figure 6C*). At *Impact*, the overall DNA methylation pattern is unchanged (*Figure 6D*), but there is a dramatic change in expression. *Impact* is normally expressed exclusively from the paternal allele, but in both MEP1 and 2, *Impact* is now expressed predominantly from the maternal allele (*Figure 6F*). In addition, in both MEP1 and MEP2, there is a significant decrease in the overall expression level of *Impact* (*Figure 6E*).

In addition to the altered imprinting at *Zac1* and *Impact*, we observe a general disruption of several additional imprinted genes. At two of these additional imprinted loci (*H19* and *Igf2r*), DNA methylation is altered (*Figure 6G* and *Figure 6—figure supplement 1A*). For example, at *H19* DNA methylation is inappropriately lost on the paternal allele in MEP1 (*Figure 6G*). There is also a slight acquisition of inappropriate methylation on the maternal allele of both MEP (*Figure 6G*). At two additional imprinted genes (*Mest* and *Snrpn*), DNA methylation appears normal (*Figure 6—figure supplement 1C,E*). Furthermore, at three of the additional imprinted loci, we observe an overall decrease in gene expression. These include *Igf2r, Mest*, and *Snrpn* (*Figure 6—figure supplement 1B,D,F*), while at *H19*, expression levels are unaffected (*Figure 6H*). However, the allele-specific expression of all of these additional imprinted loci remains unaffected (data not shown). Taken together, the changes in DNA methylation and expression indicate a general disruption in the normal regulation of genomic imprinting.

## Discussion

A major question in the field of epigenetics is whether histone methylation is regulated between generations, and whether potential defects in this process can lead to abnormalities in the resulting progeny. To address this question, we asked if the maternal reprogramming function of KDM1A, first demonstrated in *C. elegans* (*Katz et al., 2009*), is conserved in mammals. As in *C. elegans*, KDM1A is heavily deposited maternally in the mouse oocyte, suggesting that KDM1A could play a conserved role in maternal reprogramming (*Figure 1*). To address this hypothesis directly, we conditionally deleted *Kdm1a* maternally in mouse oocytes using three separate *Cre* drivers. Deletion of *Kdm1a* in developing oocytes via *Gdf9-Cre* or *Zp3-Cre* completely eliminates maternal KDM1A and results in embryonic arrest prior to the blastocyst stage, with the vast majority of embryos never developing past the 2C stage (*Figures 1,2*). This result is also observed when a second conditional *Kdm1a* allele (*Zhu et al., 2014*) is deleted with *Zp3-Cre* in the accompanying paper by Ancelin et al.. Thus, maternal KDM1A is essential for the progression of mammalian embryogenesis.

The observed 1-2C embryonic arrest does not rule out an additional role for KDM1A in oocyte maturation, as we observe that a small fraction of KDM1A deficient oocytes are unfertilized, fertilized with multiple sperm, or display other gross morphological defects, including loss of the zona pellucida. These occasional defects are also observed in the accompanying paper by Ancelin et al. Consistent with this, Kim et al. have recently reported that deletion of a different conditional *Kdm1a* allele with *Zp3-Cre* results in defects in meiotic progression (*Kim et al., 2015*). However, in contrast to our results, Kim et al. find that most oocytes are affected and are not fertilized (*Kim et al., 2015*).

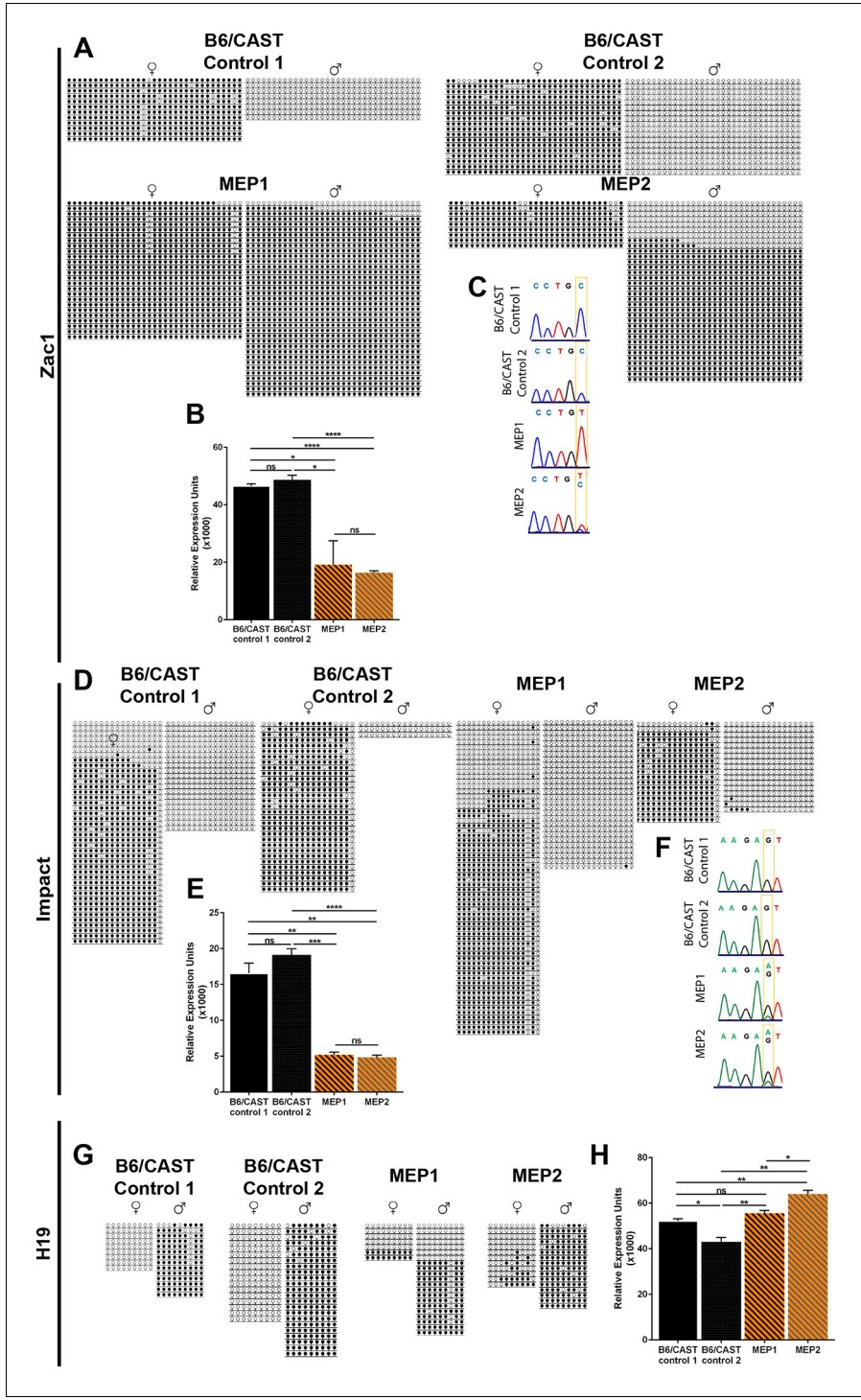

**Figure 6.** Imprinting defects in *Kdm1a^Vasa* progeny. (**A,D,G**) Allele-specific bisulfite analysis of *Zac1* (**A**), *Impact* (**D**), and *H19* (**G**). Each line represents the clone of an allele. Each circle represents a CpG dinucleotide where closed circles indicate methylation and open circles indicate no methylation. Maternal and paternal alleles are indicated. (**B,E,H**) Relative expression analysis of *Zac1* (**B**), *Impact* (**E**), and *H19* (**H**). Expression normalized to *β-actin*. Error bars indicate S.E.M. p-values calculated using an unpaired t-test with n.s. indicating p>0.05, * = p<0.05, ** = p<0.005, **** = p<0.0001. All asterisks indicate statistical significance. (**C,F**) Allele-specific expression of *Zac1* (**C**) and *Impact* (**F**). The polymorphic base is highlighted in yellow. For *Zac1*, the maternal allele SNP is T (red) in highlighted position and paternal allele SNP is C (blue) in electrophoretogram. For *Impact*, the maternal allele SNP is A (green) in highlighted position and paternal allele SNP is G (black) in electrophoretogram. All analyses

*Figure 6 continued on next page*

*Figure 6 continued*

were performed on 2 staged matched B6/CAST hybrid control pups and 2 maternal effect progeny (MEP) exhibiting perinatal lethality.

The following figure supplement is available for figure 6:

**Figure supplement 1.** Imprinting analysis of *Kdm1a^Vasa* progeny.

It is not entirely clear why the phenotypes are slightly different, but it could reflect subtle differences in the timing of deletion due to the use of different floxed alleles, or differences in strain background. Nevertheless, in *Kdm1a^Gdf9* and *Kdm1a^Zp3* mutants, we find that the majority of oocytes are fertilized normally, and many embryos undergo cleavage to reach the 2C stage (*Figure 2*). Also, in hypomorphic *Kdm1a^Vasa* animals we observe defects on the paternal allele of imprinted loci (*Figure 6*). These results suggest a requirement for KDM1A post-fertilization in the zygote. In mammals, zygotic transcription initiates between the one and two cell stage (*Aoki et al., 1997*; *Hamatani et al., 2004*; *Xue et al., 2013*). As a result, the heterozygous offspring derived from our crosses likely begin to express KDM1A from paternal allele at the 2C stage. Taken together, these results pinpoint a requirement for maternal KDM1A reprogramming between fertilization and cleavage. Consistent with this conclusion, the accompanying paper by Ancelin et al. demonstrates that chemically inhibiting KDM1A function post-fertilization also results in a 2C arrest. This further suggests a zygotic requirement for KDM1A immediately post-fertilization.

When *Kdm1a* is deleted with *Gdf9-Cre*, only one embryo reached the 2C stage. In contrast, when *Kdm1a* is deleted with *Zp3-Cre*, many embryos cleave to the 2C stage and we even observe rare embryos that reach the 3C and 4C stage (*Figure 2*). This slight difference in phenotype is likely due to the fact that the *Gdf9-Cre* acts earlier than the *Zp3-Cre*. Nevertheless, the difference in phenotype suggests that KDM1A may act maternally at multiple discrete steps. This conclusion is reinforced by the recent demonstration that loss of KDM1A also impairs meiotic progression (*Kim et al., 2015*). Importantly, the observation that many maternally deleted embryos undergo cleavage, suggests that KDM1A is not just required for cell-cycle progression or cell viability (*Figure 2*). This conclusion is consistent with the observation that KDM1A null embryos survive until e8 (*Wang et al., 2007*). Furthermore, the difference in phenotype between maternally deleted embryos and embryonically deleted embryos, which survive to e8, clearly delineates the maternal requirement for KDM1A from the embryonic requirement.

To determine the potential cause of the KDM1A maternal embryonic arrest, we profiled the expression of *Kdm1a^fl/fl* oocytes and *Kdm1a^Zp3* M-Z+ embryos that underwent cleavage to the 2C stage. Compared to control 2C embryos that have robustly activated the zygotic transcriptional program, *Kdm1a^Zp3* M-Z 2C embryos retain the expression of nearly 2 out of 3 oocyte genes, and fail to activate approximately 4 out of 5 zygotically activated embryonic genes (*Figure 3*). This indicates that maternally deleted *Kdm1a* embryos fail to undergo the MZT. Consistent with this being the primary defect in *Kdm1a^Zp3* mutants, we observe relatively few transcriptional changes in *Kdm1a^fl/fl* oocytes (*Figure 3*). The requirement for KDM1A in facilitating the MZT is also demonstrated in the accompanying paper by Ancelin et al., where it is shown that maternal KDM1A is necessary for the multiple waves of zygotic gene activation. Thus, we propose that KDM1A plays a major role in facilitating the progression of embryogenesis by enabling the MZT.

Surprisingly, in *Kdm1a^Vasa* oocytes, we found an incomplete loss of KDM1A protein, despite the fact that 100% of the progeny inherited the deleted *Kdm1a* allele (*Figure 1*). We have previously observed that strain background differences and/or specific interactions between *Cre* drivers and certain floxed alleles can lead to different patterns of deletion, often not matching published patterns. Thus, it is possible that *Kdm1a^Vasa* oocytes have a small amount of KDM1A remaining because the *Vasa-Cre* transgene occasionally works much later than previously reported when crossed with the *Kdm1a* floxed allele.

Similar to *Kdm1a^Zp3* and *Kdm1a^Gdf9* M-Z+ embryos, many of the hypomorphic *Kdm1a^Vasa* M-Z+ embryos undergo early embryonic arrest (*Figure 4*). However, the partial loss of KDM1A maternal reprogramming also uncovers a hypomorphic phenotype where a few embryos survive past the 2C stage until birth (*Figures 4–6*). Of these animals that are born, a significantly increased fraction die

perinatally (*Figure 4*). Furthermore, on the rare occasions that more than one animal was born from *Kdm1a*<sup>Vasa</sup> mutant mothers, the full litter either all died perinatally or all survived. Remarkably, the perinatal lethal pups are derived from crosses between *Kdm1a*<sup>Vasa</sup> mothers and wild-type fathers (M-Z+). This suggests that the increased perinatal lethality is due to hypomorphic maternal KDM1A in the mother's oocyte. It remains possible that the small *Kdm1a*<sup>Vasa</sup> M-Z+ litter sizes also contribute to the perinatal lethality, due to maternal neglect. However, many *Kdm1a*<sup>Vasa</sup> M-Z+ animals that are born in small litters survive to adulthood (*Table 1*). This suggests that the perinatal lethality is not simply due to small litter size.

It is possible that subtle defects in maternal KDM1A reprogramming could also lead to behavioral abnormalities in the surviving adults. To investigate this possibility, we performed a number of behavioral assays on surviving *Kdm1a*<sup>Vasa</sup> M-Z+ adult animals. These include monitoring food disappearance and bedding height to measure excessive food grinding, the marble burying assay to measure excessive digging behavior, and the open-field test, to measure anxiety. In all of these assays, we observe a maternal effect on behavior due to hypomorphic maternal KDM1A from *Kdm1a*<sup>Vasa</sup> mothers (*Figure 5*). This suggests that hypomorphic KDM1A in the mother's oocyte can lead to altered behavior in the resulting adult progeny.

To investigate the potential molecular nature of the heritable defects, we analyzed DNA methylation and expression at imprinted loci. Both DNA methylation and expression are altered at imprinted genes in perinatal lethal *Kdm1a*<sup>Vasa</sup> pups. In particular, we observe a striking disruption in the allele-specific expression of both *Zac1* and *Impact*, along with decreased expression and altered DNA methylation at other imprinted loci (*Figure 6*). This suggests that subtle defects in maternal KDM1A can alter the epigenetic landscape in a heritable fashion. Also, though KDM1A and KDM1B may have some functional overlap, the distinct imprinting defects observed in each mutant demonstrate that they have functions that are not redundant (*Ciccone et al., 2009*; *Stewart et al., 2015*). Interestingly, both *Zac1* and *Impact* have been associated with nervous system defects, and loss of *Zac1* paternally results in perinatal lethality (*Chung et al., 2011*; *Kosaki et al., 2001*; *Pereira et al., 2005*; *Roffe et al., 2013*; *Schmidt-Edelkraut et al., 2014*; *Varrault et al., 2006*). Thus, it is tempting to speculate that defects in these genes could potentially contribute to the abnormalities that we observe in *Kdm1a*<sup>Vasa</sup> M-Z+ progeny. Nevertheless, it remains unclear if this is the case.

Loss of maternal KDM1A could lead to the observed phenomena (imprinting defects, perinatal lethality and behavioral abnormalities) either directly or indirectly. Evidence for an indirect mechanism can be inferred from the misregulation of several genes (*Dnmt1, Tet1, Trim28, Zfp57, Uhrf1 and Dppa/Stella*) that are known to regulate epigenetic information. For example, *Tet1/Tet2* double mutants exhibit a partial perinatal lethality phenotype that is reminiscent of what we observe in *Kdm1a*<sup>Vasa</sup> progeny (*Dawlaty et al., 2013*). Also, mice that are haploinsufficient for *Trim28* display abnormal exploratory behaviors (*Whitelaw et al., 2010*). Alternatively, KDM1A may affect these phenomena by directly regulating histone methylation. For example, the removal of H3K4 methylation may be required to enable the activity of the DNMT3a/3b *de novo* methyltransferase complex (*Ooi et al., 2007*). Such a model has been proposed to account for the imprinting defects that are observed when KDM1B is maternally deleted (*Ciccone et al., 2009*; *Stewart et al., 2015*). Consistent with this model, the accompanying paper by Ancelin et al. demonstrates that loss of KDM1A maternally leads to global changes in both H3K4 and H3K9 methylation following fertilization. However, Stewart et al. recently demonstrated that loss of maternal KDM1A in mice does not affect H3K4me2, or the acquisition of DNA methylation at imprinted genes (*Stewart et al., 2015*). Moreover, the variable disruption of imprinting that we observe in hypomorphic animals is more consistent with a general disruption of epigenetic information. Thus, we favor a model in which the general disruption of epigenetic reprogramming, along with secondary epigenetic defects due to the misregulation of epigenetic reprogramming enzymes during the MZT, combine to give rise to the phenomena that manifest postnatally in progeny. Currently, we are engineering a hypomorphic maternal *Kdm1a* allele that can be used to generate larger numbers of hypomorphic animals to further elucidate these mechanisms.

In summary, our data highlights three important consequences of the loss of maternal KDM1A: (1) early embryonic arrest and the failure of embryos to undergo the MZT, (2) perinatal lethality and altered behavior in rare surviving progeny from hypomorphic mothers, and (3) a general disruption of genomic imprinting in the hypomorphic progeny (*Figure 7*). Although these individual phenotypes may be linked, it is possible that they are separate outcomes that depend on distinct activities

**Table 1.** Litter sizes for *Kdm1a*$^{Vasa}$ animals *Kdm1a*$^{Vasa}$ litters for mutant and control mothers indicating number of pups that died and entire litters that died. *Kdm1a*$^{Vasa}$ adult, B6/Cast control, and F2 intercrossed control litter sizes that were assayed for behavioral defects.

| | *Kdm1a*$^{Vasa}$ M+Z+ | *Kdm1a*$^{Vasa}$ M-Z+ | B6/Cast M+Z+ controls | *Kdm1a*$^{Vasa}$ M-Z+ adults | F2 intercrossed M+Z+ adults |
|---|---|---|---|---|---|
| Litter size 1 | 8 (1) | 1* | 7 | 2 | 7 |
| Litter size 2 | 7 | 4 | 3 | 1 | |
| Litter size 3 | 7 | 2 | | 4 | |
| Litter size 4 | 7 | 4 | | | |
| Litter size 5 | 7 | 1* | | | |
| Litter size 6 | 5 | 4 | | | |
| Litter size 7 | 9 | 1* | | | |
| Litter size 8 | 8 (1) | 5 | | | |
| Litter size 9 | 11 | 2* | | | |
| Litter size 10 | 8 | 2* | | | |
| Litter size 11 | 5 (2) | 2* | | | |
| Litter size 12 | 10 | 4 | | | |
| Litter size 13 | 3 | 3 | | | |
| Litter size 14 | 9 | 1* | | | |
| Litter size 15 | 5 | 3 | | | |
| Litter size 16 | 6 | 1* | | | |
| Litter size 17 | 4 | 3 | | | |
| Litter size 18 | 3 | 1* | | | |
| Litter size 19 | 11 | 1* | | | |
| Litter size 20 | 8 | 2 | | | |
| Litter size 21 | 2* | | | | |
| Litter size 22 | 7 | | | | |
| Litter size 23 | 10 | | | | |
| Litter size 24 | 8 | | | | |
| Litter size 25 | 8 | | | | |
| Litter size 26 | 3 (1) | | | | |
| Litter size 27 | 9 (1) | | | | |
| Litter size 28 | 11 | | | | |
| Litter size 29 | 7 | | | | |
| Litter size 30 | 4 | | | | |
| Litter size 31 | 5 (1) | | | | |
| Litter size 32 | 1 | | | | |

( ) indicates pups that died.

* indicates entire litter died.

of maternal KDM1A during the MZT. Nevertheless, our results establish a novel mammalian paradigm in which altered epigenetic reprogramming by a histone demethylase between generations leads to defects that manifest weeks later, highlighting the potential long-range consequences of epigenetic perturbations in the early mammalian embryo.

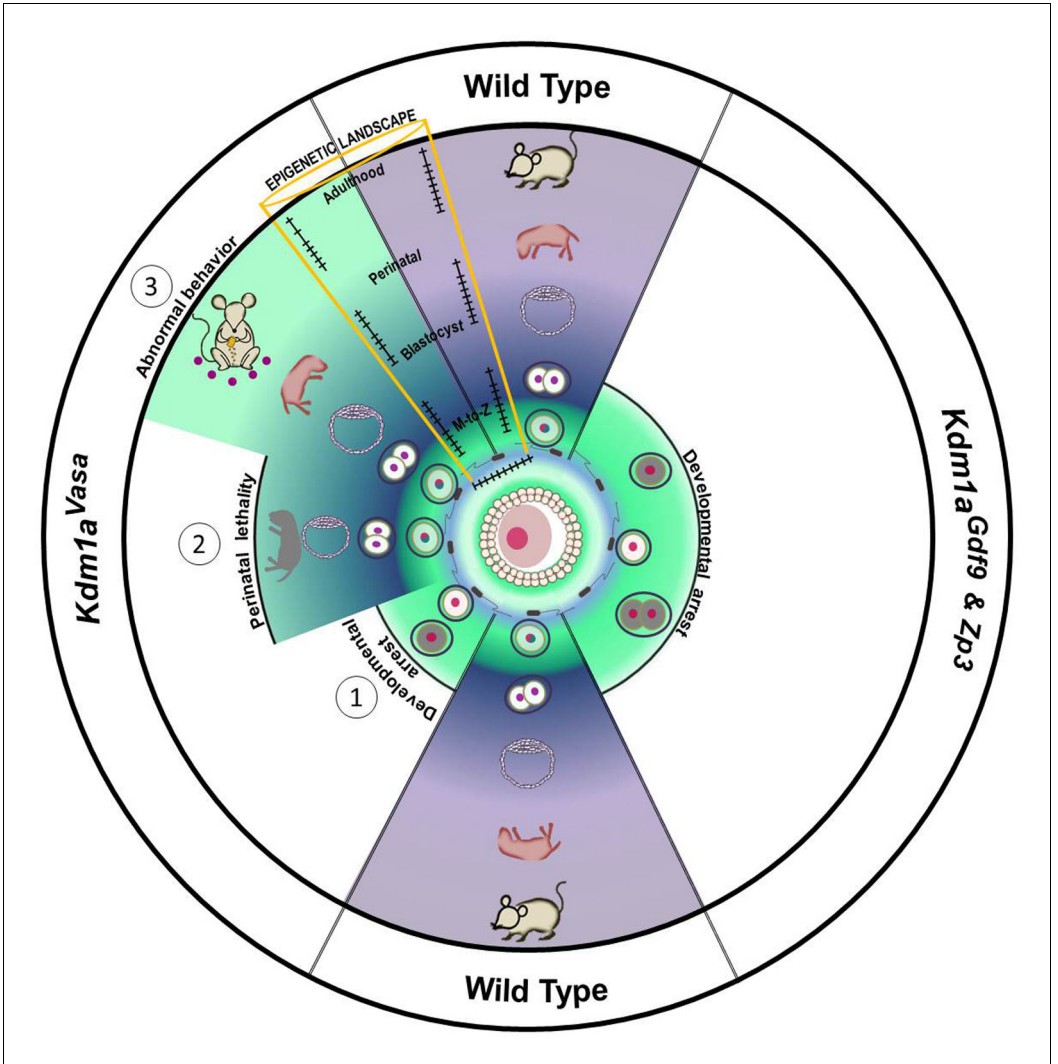

**Figure 7.** Model. Loss of maternal LSD1 results in defects later in development in wild-type oocytes, after fertilization (denoted by blue sperm encircling oocyte) the fertilized egg undergoes the maternal to zygotic transition (MZT; green to blue/purple) at the 1–2 cell stage. These M+Z+ embryos proceed normally through development (indicated by blastocyst, perinatal stage pup, and adult mouse). In contrast, when Lsd1 is deleted with either Gdf9- or Zp3-Cre, the resulting Lsd1Gdf9 and Lsd1Zp3 progeny become arrest at the 1–2 cell stage and never undergo the MZT (green). When Lsd1 is deleted with Vasa-Cre, we observe 3 hypomorphic outcomes in resulting Lsd1Vasa progeny: (1) developmental arrest at the 1–2 cell stage, (2) perinatal lethality and (3) abnormal behavior in surviving adult animals. These outcomes are due to reduced LSD1 in the mothers oocyte, suggesting that lowered maternal LSD1 can result in defects much later in development. These long-range outcomes are associated with imprinting defects (depicted as wild-type versus mutant changes in DNA methylation within the yellow region).

# Materials and methods

## Generation of KDM1A oocyte-specific mutant females

The following mouse strains were used: *Kdm1a/Lsd1^{fl/fl}* MGI: 3711205 (*Wang et al., 2007*), *Gdf9-Cre* MGI: 3056522 (*Lan et al., 2004*), *Ddx4/Vasa-Cre* MGI: 3757577 (*Gallardo et al., 2007*), and *Zp3-Cre* MGI: 2176187 (*de Vries et al., 2000*) animals. To generate *Kdm1a* oocyte conditional knockout mice, *Kdm1a/Lsd1^{fl/fl}* females were crossed to transgenic *Cre* males to generate *Kdm1a/*

$Lsd1^{\Delta/+}$ F1 animals with *Cre*. F1 males were crossed to transgenic *Cre* females to generate *Kdm1a/Lsd1*$^{\Delta/+}$ F2 animals with *Cre* and F2 males were mated to *Kdm1a/Lsd1*$^{fl/fl}$ females to generate *Kdm1a/Lsd1*$^{\Delta/+}$ F2 animals with *Cre*. These maternally deleted *Kdm1a/Lsd1*$^{\Delta/\Delta}$ females were then mated to wild-type B6 or *M. castaneus* males to produce maternal effect progeny. Prior to the initial crosses, *Kdm1a/Lsd1*$^{fl/fl}$ were mated to C57BL/6 mice for several generations so that the genetic background is mostly B6. For *Gdf9-Cre* and *Zp3-Cre, Kdm1a/Lsd1*$^{\Delta/+}$ F1 animals with *Cre* were mated directly with *Kdm1a/Lsd1*$^{fl/fl}$ females. A table is included with the litter sizes of all perinatal lethal animals and all animals assayed in the behavioral assays (See *Table 1*). All mouse work was performed under the approved guidelines of the Emory University, NIH and Salk Institute IACUC. Polymorphisms were identified through the UCSC Genome Browser and subsequently verified through sequencing of parental strains and hybrid progeny (See *Table 2*).

**Table 2.** Allele-specific primers and polymorphisms, each primer and polymorphism used for allele-specific analysis bisulfite analysis and allele-specific expression analysis.

| Gene name | Use (Bisulfite or qPCR) | Verified SNP (rs#) | Primers |
|---|---|---|---|
| Zac1 | Allele-Specific Bisulfite Analysis | A-B6/ G-Cast (rs29364824) | For- G GGTAGGTAAGTAGTGATAA |
| | | | Rev- C CTAAAACACCAAAATAACA |
| | qPCR | T-B6/C-Cast (rs33583472) | For- CATTTGTAGGCATGCCCGTC |
| | | | Rev- G TGGTAGCTGCATCTGGGGCTGGA |
| Impact | Allele-Specific Bisulfite Analysis | A-B6/G-Cast (rs31057356) | For- TTGTATAGTTTTGTTTTTATAAGTG |
| | | | Rev- AACCTACTCATATAACAATACAAC |
| | qPCR | A-B6/ G-Cast (rs31052361) | For- GAAGAAAACTGAAGAGGTTG |
| | | | Rev- GCATAGATGTTGTGGGTGGC |
| H19 | Allele-Specific Bisulfite Analysis | G-B6/ A-Cast (verified from Bartolomei Lab) | For- ATTTATAAATGGTAATGTTGTGG |
| | | | Rev- CCTCATAAAACCCATAACTATAAAATC |
| | qPCR | | For- CCACTACACTACCTGCCTCAGAATCTGC |
| | | | Rev- GGTGGGTACTGGGGCAGCATTG |
| Igf2r | Allele-Specific Bisulfite Analysis | G-B6/ A-Cast (rs107811421) | For- TAGAGGATTTTAGTATAATTTTAA |
| | | | Rev- TAACACTTTTAAATTCATCTCT |
| | qPCR | | For- CTGGAGGTGATGAGTGTAGCTCTGGC |
| | | | Rev- GAGTGACGAGCCAACACAGACAGGTC |
| Mest | Allele-Specific Bisulfite Analysis | G-B6/ T-Cast (rs245841095) | ForMat- GGGTGTTTTATGTTTTTTAGGGT; ForPat- GGGTGTTTTATGTTTTTTAGGGG |
| | | | Rev- CCCAAATTCTAATAAAAAAAACCTTCCCAT |
| | qPCR | | For- GCTGGGGAAGTAGCTCAGT |
| | | | Rev- TTTCTTCTTAGCAAGGGCCA |
| Snrpn | Allele-Specific Bisulfite Analysis | A-B6/T-Cast (rs50790468) | ForMat- GTAATTATATTTATTATTTTAGATTGATAGTGAT; ForPat- GTAATT ATATTTATTATTTTAGATTGATAGTGAG |
| | | | Rev- ATAAAATACACTTTCACTACTAAAATCC |
| | qPCR | | For- TGC TCGTGTTGCTGCTACTG |
| | | | Rev- GCAGTAAGAGGGGTCAAAAGC |
| β-actin | qPCR | | For- G TGACGAGGCCCAGAGCAAGAG |
| | | | Rev- C GTACATGGCTGGGGTGTTGAAGG |

## Isolation of pre-implantation mouse embryos

To establish phenotypes in KDM1A maternal mutants, timed matings were set up between control/mutant females and wild-type males. Superovulation was found to have little affect on phenotypes, so we used superovulation to collect enough embryos for RNA-seq. For super-ovulation, PMSG was injected into sexually mature females on day 1. After 48 hr, females were injected with HcG and subsequently housed with stud males. Confirmation of natural or superovulatory matings on subsequent days was made via observation of a copulation plug. 1–2 cell embryos were flushed from ovarian tract at embryonic day 1.5 (e1.5), morulae on e2.5 and blastocysts on e3.5. Flushed embryos were categorized and imaged.

## Immunofluorescence

Isolated mouse ovaries were fixed for 1 hr in 4% PFA on ice then washed with PBS multiple times over a 2 hr period. Ovaries were allowed to sit in 30% sucrose solution at 4°C overnight and were then embedded in OCT compound. 10 micron cryosections were obtained for analysis. Immunostaining was performed using rabbit polyclonal anti-KDM1A (1:200, Abcam, Cambridge UK, ab17721) and Alexa fluor conjugated secondary antibodies.

## Immunohistochemistry

Isolated mouse ovaries were fixed overnight in 4% PFA. Ovaries were then dehydrated in the following series of steps: 70% ethanol for 20 min 3 times; 85% ethanol for 45 min 2 times; 95% ethanol for 1 hr; 100% for 1 hr; xylenes overnight; xylenes:paraffin mix for 2 hr twice under vacuum; paraffin for 4 hr under vacuum; and paraffin under vacuum overnight. Oocytes were embedded in paraffin and 10 micron sections were taken for analysis. Immunostaining was performed using rabbit polyclonal anti-KDM1A (1:500, Abcam, ab17721). Oocytes were scored for presence of KDM1A signal with qualitative comparison to wild-type oocyte signal.

## Bisulfite analysis

DNA was isolated from sagittal sections of each perinatal pup. Bisulfite conversion was performed according to the Zymo (Irvine, CA) EZ DNA Methylation Kit protocol. Primer sets and polymorphisms used are listed in *Table 1*. The BiQ Analyzer program was used in the analysis of bisulfite converted sequences (*Bock et al., 2005*).

## Quantitative real-time PCR analysis

RNA was isolated from sagittal sections of each perinatal pup using Trizol. SuperScript® III first-strand synthesis system was used to generate cDNA. The following cycling conditions were used: 95°C for 3 min, 95°C for 15 sec, 60°C for 30 sec, 72°C for 30 sec, 50 cycles.

## Genome-wide expression analysis

RNA-seq on KDM1A mutant and wild-type 2C embryos was performed as described (*Macfarlan et al., 2012*). Briefly embryos were lysed in Prelude Direct Lysis buffer (Nugen, San Carlos, CA) and amplified cDNAs were prepared using the Ovation RNA-seq system V2 (Nugen, 7102-32). Paired end libraries were prepared according to the Tru-seq library construction protocol starting with Covaris fragmentation step. Libraries were then sequenced on an Illumina Hi-seq 2000. For expression analysis, only first mate pair was used and reads were trimmed from 3' end to 50 bp. Trimmed reads were filtered to a minimum average base quality of 15. We combined the known-Gene, ensGene and refGene annotations for mm10 (downloaded from UCSC Genome Browser) with the full RepeatMasker annotation, also from UCSC Genome Browser, to build a single gene annotation. Redundant transcripts were filtered out using the gffread utility packaged with Cufflinks. The annotation was modified to include a common gene_id value for same-name repeat elements resulting in a total of 875 repeats. Sequences for all annotated features were extracted from the mouse genome and RNA-Seq reads were aligned to them using BBMap with 95% identity, up to a single INDEL and up to 2000 equally best alignments per read. Alignments to features were quantified by counting hits to gene loci and down-weighting read alignments that mapped to multiple gene loci by $1/N^2$ where N is equal to the number of gene loci. Reads aligned to repeat elements were

counted in the same way but instead of using gene loci the repeat name was used for binning hits. If a read mapped to multiple repeat features all of the same name then it was counted as 1 hit to that repeat name. Raw read counts were loaded into R and median normalized for differential expression analysis. For each pair-wise test genes with raw counts less than 10 in all conditions were not tested. Differential expression was performed by testing the null hypothesis that a gene's fold change between conditions is zero. The observed dispersion-mean relationship was fit using a method similar to that of DESeq and the predicted dispersions were used as the minimum dispersion per gene. Fold-changes between conditions were tested by monte-carlo simulation using the observed and estimated means and dispersions to generate random negative binomial distributed count values (the rnbinom method in R). In each monte-carlo iteration N samples were simulated per condition (N = to the number of samples in each condition), averaged into conditions and the log fold-change was calculated. The resulting simulated log fold-change distributions are normally distributed. Z-scores were computed for each observed log fold-change compared to its corresponding monte-carlo simulated fold-change distribution and translated to two-tail p-values against the normal probability distribution. Raw p-values were adjusted using the Benjamini & Hochberg correction. Genes and repeats were tested together and marked significant if the adjusted p-value was less than 0.05. RNA-seq data is deposited at GEO (GSE66547). Oocyte expression data used for comparisons was previously deposited at GEO (GSE33923.) For the heat map plot, each gene found to be significantly associated with PC1 is standardized relative to its mean and scaled relative to its variance (ie each gene is scaled separately to improve visualization.) Each row is a gene which is standardized where mean is zero with standard deviation is 1. This makes each row relative to its mean. The scale is relative to its variance. Rows are sorted by a clustering of the genes. Color scale is relative to mean expression level per gene.

## Characterization of food-grinding behavior in $Kdm1a^{Vasa}$ adult progeny

Each animal was placed in a mouse housing unit (32.8 × 18.6 × 13.6 cm) with 5/8 inches of bedding. Over an 6 day period, the height of the bedding was measured in inches. In addition, 400 grams of standard mouse diet pellets was placed in each food hopper. The amount of food remaining in the food hopper was weighed each day over a 3 day period.

## Marble burying assay

A clean, transparent plastic cage (32.8 × 18.6 × 13.6 cm) was filled 4.5 cm deep with corncob bedding material. 20 glass marbles (20 mm diameter) were evenly spaced in a 5 × 4 grid on the surface of the bedding. During the testing phase, the mice were placed in the cage for 25 min and allowed to explore. At the end of the testing phase, mice were removed from the cage and the number of marbles that were buried 2/3 their height in the bedding were counted.

## Open field test

Mice were placed in a clean, transparent plastic cage (55.5 × 32.5 × 19.5 cm) with grid lines marked on bottom of cage. The center area was clearly marked for analysis (27 × 16.2 cm). After 10 min, video was scored for number of times center area was crossed and amount of time spent in center area.

## Acknowledgements

We thank the Transgenic Mouse and Gene Targeting Core Facility (TMF) specifically T. Caspary, T. Quackenbush, K. Piotroska-Nitsche and H. Zhang for their assistance with the superovulation and embryo isolation experiments. Thank you to the Emory DAR for providing animal husbandry, the Dept. of Cell Biology for a wonderful and scientifically stimulating environment, the BCDB program for their continued support and the entire Katz Lab, specifically Michael Christopher and Teresa Lee, as well as T. Caspary for helpful discussions and comments on the manuscript. Thank you to M. Rosenfeld for providing the $Kdm1a^{fl//fl}$ mice, D. Castrillon for providing the Vasa-Cre mice, M. Bartolomei for information on verified H19 polymorphisms and primers, S. Gourley and J. Schroeder (the Emory Rodent Behavioral Core) for helpful advice on mouse behavioral assays. Thank you to A.K. and S.K. for providing the marbles. Gdf9-Cre and Zp3-Cre mice were obtained from the Jackson

Laboratories (Bar Harbor, ME). J.A.W was supported by the Biochemistry, Cell and Molecular Biology Training Grant (5T32GM008367) and D.A.M by the Emory PREP Post-Bac Program (5R25GM089615-04). The work was supported by a grant to D.J.K from the National Science Foundation (IOS1354998). T.S.M. is supported by Eunice Kennedy Shriver National Institute of Child Health and Human Development DIR grant HD008933.

## Additional information

### Funding

| Funder | Grant reference number | Author |
|---|---|---|
| National Science Foundation | IOS 1354998 | Jadiel A Wasson David J Katz |

The funders had no role in study design, data collection and interpretation, or the decision to submit the work for publication.

### Author contributions

JAW, TSM, Conception and design, Acquisition of data, Analysis and interpretation of data, Drafting or revising the article; AKS, Conception and design, Acquisition of data, Analysis and interpretation of data; DAM, GW, Acquisition of data; SD, Analysis and interpretation of data; SLP, Conception and design, Analysis and interpretation of data; DJK, Conception and design, Acquisition of data, Analysis and interpretation of data, Drafting or revising the article, Contributed unpublished essential data or reagents

### Author ORCIDs

David J Katz, http://orcid.org/0000-0002-3040-1142

### Ethics

Animal experimentation: All mouse work was performed under the approved guidelines of the Emory University IACUC (protocol #2002534).

## Additional files

### Major datasets

The following datasets were generated:

| Author(s) | Year | Dataset title | Dataset URL | Database, license, and accessibility information |
|---|---|---|---|---|
| Todd S Macfarlan | 2015 | RNA-seq of Maternal Kdm1a mutant 2C embryos and eggs | http://www.ncbi.nlm.nih.gov/geo/query/acc.cgi?acc=GSE66547 | Publicly available at the NCBI Gene Expression Omnibus (Accession no: GSE66547). |

The following previously published dataset was used:

| Author(s) | Year | Dataset title | Dataset URL | Database, license, and accessibility information |
|---|---|---|---|---|
| Todd S Macfarlan, Gifford WD, Shawn Driscoll, Lettieri K, Rowe HM, Bonanomi D, Firth A, Singer O, Trono D, Samuel L Pfaff | 2012 | 2C::tomato ES cells, 2-cell embryos and wild type oocytes | http://www.ncbi.nlm.nih.gov/geo/query/acc.cgi?acc=GSE33923 | Publicly available at the NCBI Gene Expression Omnibus (Accession no: GSE33923). |

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
