## [Decision Letter]

Thank you for submitting your work entitled "Maternally provided LSD1 enables the maternal-to-zygotic transition and prevents defects that manifest postnatally" for peer review at *eLife*. Your submission has been favorably evaluated by Janet Rossant (Senior editor), a Reviewing editor, and two reviewers, one of whom, Gavin Kelsey, has agreed to share his identity.

All agree that the work represents a significant advance however several revisions are recommended. The Reviewing editor has drafted a summary of the reviews to help you prepare a revised submission.

In this study, the authors nicely show that deletion of LSD1/KDM1A in oocytes in the mouse (which they do with three different *Cre*-drivers) causes arrest at the 2-cell stage and gross impairment of zygotic genome activation (as demonstrated by a nicely executed RNA-seq analysis). The mechanistic basis of these effects are not characterised in further detail. The absolute requirement of LSD1 for embryonic progression, implicating the importance of regulation of histone post-translation modifications, is an important observation. The unique aspect of this manuscript, however, is the *Lsd1* hypomorph inadvertently produced by the incomplete ablation caused by the *Vasa-Cre* (paradoxical given the earlier timing of this *Cre* transgene and remains unexplained). The imperfect ablation of *Lsd1* allows some embryos to survive through the critical pre-implantation phase, and then to term and adulthood, although there is another period of attrition at birth. The adult survivors display abnormal behaviours, which are characterised in detail. Again, I think this is a very interesting observation, which points to epigenetic imbalance in the early embryo as having potentially long-term consequences. No doubt, the authors could have chosen other phenotypes to examine, as there is no reason to suspect that the presumed epigenetic defects should be restricted to these few behaviours, but the limited numbers of surviving adults preclude a deeper phenotypic characterisation. With justification, the authors choose to analyse imprinted genes as potential mediators, although I am not aware that the specific imprinted genes examined are implicated in the behaviours tested.

Specific comments:

1) I have a problem with the basic experimental design. It seems to me that if one expects developmental failures one should never use superovulated females as a source of the eggs. Eggs and embryos from superovulated females are going to be inferior and any negative developmental effect will be potentiated. I would recommend that at least some of described experiment be repeated using normal matings. In addition, the only significant molecular information in the manuscript is the transcriptome comparison between the maternally deleted and wt 2-cell stage embryos i.e. period of zygotic genome activation (ZGA). However, the ZGA is entirely regulated by the maternally inherited molecules; proteins and RNAs. Proteome comparison would be technically quite difficult but transcriptome comparison between maternally deleted and wt oocyte is possible, necessary and should be included and interpreted in the revised manuscript. Furthermore, cytoplasmic polyadenylation is one of the most important mechanisms controlling the utilization of many maternal mRNAs and it would be very important to determine how it functions during maturation of mutant oocytes and after fertilization in zygotes. In order to supplement the data on the imprinting state of several genes in perinatal pups, it would be necessary to assess and include in the manuscript the imprinting state of the same genes in the ovulated oocytes and zygotes.

2) A major concern is the characterisation of the effects on methylation and expression of imprinted genes in the *Lsd1^Vasa^* progeny; this should be improved, especially because the link between effects on methylation and expression do not appear to be well correlated. I think there are obvious ways to improve this and the analysis as presented does not allow strong conclusions.

3) Is it the case that the authors analysed methylation and expression in dead pups (they refer to them as 'exhibiting perinatal lethality)? In which case, how reliable is the RT-qPCR analysis?

4) Do the bisulphite sequencing results take into account clonality? For example, for the *Igf2r* DMR in MEP1 (Figure 6—figure supplement 1), there appears to be essentially three patterns of methylation: one fully unmethylated sequence, a set of fully methylated sequences and a set of sequences that are partially methylated in an identical pattern. Such a pattern could suggest a high rate of clonality and that very little amplifiable DNA went into the bisulphite reaction or post-bisulphite PCR, perhaps because DNA was degraded in dead pups. Alternatively, the result could suggest something of more biological interest, that these were the three methylation profiles present in cells of the early cleavage embryo at that very limited time window of maternal LSD1 deficiency, and that this limited set of profiles becomes 'fixed' when the embryonic lineage is determined from a small number of blastomeres. I think we need more detail and improved methylation analysis to be able to discriminate between technical issues and biologically interesting interpretation.

5) What tissue is actually assessed? The authors state 'sagittal sections' of the pups. So, is this comparable between the controls and mutants? Is it the case that only one control and two mutant pups were examined (the figure legend states: “All analyses were performed on a staged matched B6/CAST hybrid control pup and 2 maternal effect progeny (MEP) exhibiting perinatal lethality”)? This does not seem adequate for expression level comparison.

6) In the subsection “LSD1 is expressed throughout oocyte development”, finding of incomplete loss of LSD1 is really surprising. In the paper originally describing *Vasa-Cre* mice, deletion was always observed in all oocytes. The breeding schemes the authors are using indicate that there is only one functional allele of *Lsd1* in the target cells, which makes the persistence of LSD1 even more surprising. Considering that *Vasa-Cre* is most likely expressed in primordial germ cells it is extremely unlikely that, if deletion took place, the LSD1 can persist till ovulation. The authors admit that they cannot explain it, however the question remains if one should leave it like that. The authors can check how efficient deleters are the *Vasa-Cre* mice they are using (more about mice later) by crossing them with some other *fl/fl* target strain. In addition, it would be interesting to determine the level of LSD1 in maternally deleted eggs. Is it the same as in heterozygous females or significantly lower? This information would be very important in evaluating the results of behavioural tests.

7) In the first paragraph of the subsection “Loss of maternal LSD1 results in 1-2 cell embryonic arrest”, is the timing of expression of paternal allele known? Please provide experimental evidence or citation.

8) In the first paragraph of the subsection “Loss of maternal LSD1 results in 1-2 cell embryonic arrest”, the development of *Lsd1^Gdf9^* control embryos is very poor. Another example how unrelated/uncontrolled influence can significantly affect experimental results. In view of these results I would suggest deleting all results in which *Gdf9-Cre* is used. In the last paragraph of the aforementioned subsection, development of embryos from *Zp3-Cre* crosses is obviously much better than from *Gdf9* crosses, strain effect? Another reason to delete *Gdf9* results as those two groups should be comparable. 3- and 4-cell stage embryos appear very unhealthy.

9) One note of caution on the behavioural effects is that there could be a potential confound of litter effect, given that factors such as litter size in the early postnatal period could have a programming effect on a variety of adult behaviours – and so could be quite independent of history of *Lsd1* deficiency per se – and there is a strong effect on litter size (Figure 4). Stronger data on the effect on imprinted genes (see above) would help negate a possible confounding effect such as this.

10) From the RNA-seq datasets, the authors comment on mis-regulation of interesting DNA methylation modulators, such as *Dnmt1, Tet1, Trim28, Zfp57, Dppa3/Stella* in 2-cell embryos from *Lsd1^Zp3-^Cre* females. Ideally, these effects would be validated by RT-qPCR. Also, this mis-regulation could be more apparent than real. For example, a two-fold up-regulation of *Dnmt1* could represent the same number of oocyte-derived *Dnmt1* transcripts, but a reduction in the total mRNA pool in the 2-C embryo owing to impaired zygotic genome activation, so it is a relative rather than absolute change.

[Editors' note: further revisions were requested prior to acceptance, as described below.]

Thank you for resubmitting your work entitled "Maternally provided LSD1/KDM1A enables the maternal-to-zygotic transition and prevents defects that manifest postnatally" for further consideration at *eLife*. Your revised article has been favorably evaluated by Janet Rossant (Senior editor), a Reviewing editor, and two reviewers. The manuscript has been improved but there are some remaining issues that need to be addressed before acceptance, as outlined below. These can be addressed without further experimentation, with modifications to the manuscript text or methods.

*Reviewer #1:*

The authors addressed (or attempted to address) most of the comments. They also provided some additional experiments and discussed why some of the requested experiments could not be performed. The arguments (shortage of time, being out of the scope of the investigation, limit imposed by the available number of embryos) seem overall reasonable and justified. The manuscript is of sufficient quality and interest to justify publication in *eLife* though some additions in the text are necessary.

Additional comment:

The Discussion is mostly concentrating on relatively trivial and unexplained results dealing with imprinting errors observed in two dead pups. The authors tried to explain the reason for using dead pups and how they controlled for possible artefacts. This explanation is not entirely satisfactory and possible technical problems could explain why two samples significantly differ. I certainly do not see the need to provide so much space in illustrations, results and discussion to this subject. The rest of the Discussion is mostly a repetition of the Results. There is no attempt to speculate about some puzzling observations. The presence of a small amount of protein in supposedly *Vasa-Cre* mediated deleted oocytes has been partially explained in the authors' reply to reviewers so why not add those thought to the Discussion. The suggestion that a possible reason may be extreme stability of LSD1 protein or RNA is not logical. If this would be the case one would expect to see even more protein or RNA in the oocytes in which deletion happens much later when mediated by *Gdf9-Cre* and *ZP3-Cre*.

The authors (again in the accompanying letter) state that differences between the *Gdf9-Cre* and *ZP3-Cre* experiments may be caused by different locations. They could have easily checked if this is true. As it stands there is a very significant difference and readers may draw all kinds of conclusions from what could be the trivial fact that one lab is better in growing embryos or handling mice. However, it is also possible that the difference is caused by an earlier action of GDF9 which would in turn deprive oocytes of histone demethylase a few days earlier. However if LSD1 protein or RNA is so long lived this explanation is useless.

One would really like to see those puzzles discussed a bit better.

*Reviewer #2:*

As before, this manuscript on the consequences of ablation of LSD1/KDM1A in mouse oocytes has three components: phenotypic and molecular characterisation of defects in oocytes and preimplantation embryos; description of behavioural effects in rare surviving adult progeny obtained after early maternal germline ablation of LSD1 (*Vasa-Cre*); and molecular analysis of imprinted genes in pups from such maternal germline mutants. Some connections can be made between these phenotypic outcomes, but they can be considered separate facets of the story. Thus, the reported behavioural defects could be completely unrelated to altered imprinting, but the analysis of imprinted genes offers precedent that maternal lack of LSD1 could give rise to early epigenetic errors that persist into adult life. In the revised manuscript, the characterisation of the molecular defects in early embryos is significantly improved with the addition of the RNA-seq data from *Lsd1:Zp3-Cre* oocytes. These added data allow the authors to conclude more strongly that there is a defect in zygotic genome activation, since the disruption in gene expression in oocytes is far less extensive. However, whether the ZGA defect depends upon activity of LSD1 in the zygote/embryo or other factors stored in the oocyte that depend upon LSD1 in the oocyte cannot be distinguished. The manuscript from Ancellin et al. addresses this point. Nevertheless, the informatic analysis of the RNA-seq datasets looks very robust.

Regarding the other two main aspects of the manuscript, I retain some concerns. I think the authors could have gone about the expression and methylation analysis in a more robust way. Given the prominence given to this analysis in the manuscript, I would have considered that sacrificing a couple of pups for more refined molecular analysis, rather than relying on the approach adopted of extracting DNA and RNA from histological sections of deceased pups, would have been appropriate. It would have given this analysis far greater credibility: we would have had the confidence that pristine, freshly isolated and defined tissues were being analysed, rather than an aggregation of tissue of unknown quality from a sagittal section. On a side point, regarding the result for *Grb10* (Figure 6—figure supplement 1), it is not clear that the authors are actually looking at the germline DMR, as this region should be fully unmethylated on the paternal allele of control animals, whereas both alleles seem to be fully methylated. (The germline DMR is unmethylated on the paternal allele in all tissues and, as far as I am aware, tissue-specific expression status of *Grb10* occurs without changes in the methylation of the germline DMR.)

Regarding the behavioural tests, I am not sure the issue of litter size and behavioural effects is fully dealt with. Were the litter sizes similar for the various control groups, given the very small litter sizes of the maternal mutants? The definitive ways to get around possible confound of litter size when there seems to be such a profound effect on litter size would be to reduce the litter size of the controls groups to those of the maternal mutant class, or undertake cross-fostering to ensure control and maternal mutant groups all experience similar postnatal care. In as much as it is not possible to expect the authors to undertake the study in this way now, I do strongly urge them to include explicit statements in the Materials and methods on the postnatal history of the adults, both maternal mutant and control groups, relating to litter size (exact litter sizes for all groups). There remains a potential confounding factor here, and readers need to be able to appreciate it.

---

## [Author Response]

*Specific comments: 1) I have a problem with the basic experimental design. It seems to me that if one expects developmental failures one should never use superovulated females as a source of the eggs. Eggs and embryos from superovulated females are going to be inferior and any negative developmental effect will be potentiated. I would recommend that at least some of described experiment be repeated using normal matings.*

The *Lsd1*_Zp3_ experiments performed to identify the 1-2 cell arrest were originally performed with natural matings. Superovulation was then necessary to get more embryos for subsequent analyses. The text has been amended to more clearly reflect this. In addition, all *Lsd1*_Vasa_ experiments were performed with natural matings.

*In addition, the only significant molecular information in the manuscript is the transcriptome comparison between the maternally deleted and wt 2-cell stage embryos i.e. period of zygotic genome activation (ZGA). However, the ZGA is entirely regulated by the maternally inherited molecules; proteins and RNAs. Proteome comparison would be technically quite difficult but transcriptome comparison between maternally deleted and wt oocyte is possible, necessary and should be included and interpreted in the revised manuscript. Furthermore, cytoplasmic polyadenylation is one of the most important mechanisms controlling the utilization of many maternal mRNAs and it would be very important to determine how it functions during maturation of mutant oocytes and after fertilization in zygotes.*

We agree and have prioritized these studies. We now include RNA-seq analysis of polyadenylated and non- polyadenylated RNAs in *Lsd1*_Zp3_ mutant versus control oocytes (Figure 3). This analysis shows that the transcriptome differences between control and *Lsd1*_Zp3_ oocytes are much less than the differences between control and *Lsd1*_Zp3_ M-Z+ 2C embryos. This finding is consistent with our conclusion that LSD1 functions post-fertilization during the maternal-to-zygotic transition. The new RNA-seq analysis does have some interesting new leads, including the massive upregulation of the Metallothionein1 (*Mt1* gene) and the reduction in several mitochondrially encoded genes in *Lsd1*_Zp3_ oocytes. This indicates a possible stress response or metabolic deficit in the mutants which we have now indicated in the main text and Discussion. We are currently pursuing these potential *Lsd1*_Zp3_ oocyte defects in our ongoing studies.

*In order to supplement the data on the imprinting state of several genes in perinatal pups, it would be necessary to assess and include in the manuscript the imprinting state of the same genes in the ovulated oocytes and zygotes.*

This is a good question and was part of the original rationale for these experiments. So we tried very hard to analyze the establishment of maternal imprints in *Lsd1*_Zp3_ and *Lsd1*_Vasa_ mutants, both in oocytes and arrested embryos (in the Katz and Macfarlan Laboratories). Although we could amplify repeat sequences and detected little change in DNA methylation levels in mutants relative to wt, we have been unable to amplify ICR regions from bisulfite treated DNA, even with nested PCR. Importantly, in our new RNA-seq analysis we do not observe imprinted gene expression defects in *Lsd1*_Zp3_ oocytes. This provides indirect support that imprints are properly established.

*2) A major concern is the characterisation of the effects on methylation and expression of imprinted genes in the Lsd1^Vasa^ progeny; this should be improved, especially because the link between effects on methylation and expression do not appear to be well correlated. I think there are obvious ways to improve this and the analysis as presented does not allow strong conclusions.*

We have now assayed DNA methylation at imprinting control regions in a second control neonatal pup (Figure 6). In this second control, imprinting is even more strictly maintained, suggesting that the imprinting defects in *Lsd1*_Vasa_ pups are not just due to variability. We were unable to obtain additional surviving *Lsd1*_Vasa_ mutants to improve the analysis on the mutant side. The variability between effects on methylation and expression are consistent with our overall observations that multiple epigenetic enzymes are disrupted.

*3) Is it the case that the authors analysed methylation and expression in dead pups (they refer to them as 'exhibiting perinatal lethality)? In which case, how reliable is the RT-qPCR analysis?*

Because of the very limited number of hypomorphic animals and the need to allow viable neonatal pups to develop into adults, the methylation and expression had to be done on sagittal sections of newly deceased pups. In order to minimize the effects of death on these analyses, careful attention was taken to match the death conditions in controls. We also took great care to match sagittal sections. Importantly, we have now provided new RT-PCR data from a second control pup (Figure 6). The new data closely matches the other control, suggesting that the variability in expression that we see in some of *Lsd1*_Vasa_ pups is likely due to real variability rather than assay conditions.

*4) Do the bisulphite sequencing results take into account clonality? For example, for the Igf2r DMR in MEP1 (Figure 6—figure supplement 1), there appears to be essentially three patterns of methylation: one fully unmethylated sequence, a set of fully methylated sequences and a set of sequences that are partially methylated in an identical pattern. Such a pattern could suggest a high rate of clonality and that very little amplifiable DNA went into the bisulphite reaction or post-bisulphite PCR, perhaps because DNA was degraded in dead pups. Alternatively, the result could suggest something of more biological interest, that these were the three methylation profiles present in cells of the early cleavage embryo at that very limited time window of maternal LSD1 deficiency, and that this limited set of profiles becomes 'fixed' when the embryonic lineage is determined from a small number of blastomeres. I think we need more detail and improved methylation analysis to be able to discriminate between technical issues and biologically interesting interpretation.*

The bisulfite analysis was performed on a large amount of DNA extracted from a sagittal section of a neonatal pup. This minimizes the chances of clonality. In addition, we have verified the lack of clonality using the BiQ analyzer program.

*5) What tissue is actually assessed? The authors state 'sagittal sections' of the pups. So, is this comparable between the controls and mutants? Is it the case that only one control and two mutant pups were examined (the figure legend states: “All analyses were performed on a staged matched B6/CAST hybrid control pup and 2 maternal effect progeny (MEP) exhibiting perinatal lethality”)? This does not seem adequate for expression level comparison.*

Please see responses to points 2 and 3.

*6) In the subsection “LSD1 is expressed throughout oocyte development”, finding of incomplete loss of LSD1 is really surprising. In the paper originally describing Vasa-Cre mice, deletion was always observed in all oocytes. The breeding schemes the authors are using indicate that there is only one functional allele of Lsd1 in the target cells, which makes the persistence of LSD1 even more surprising. Considering that Vasa-Cre is most likely expressed in primordial germ cells it is extremely unlikely that, if deletion took place, the LSD1 can persist till ovulation. The authors admit that they cannot explain it, however the question remains if one should leave it like that. The authors can check how efficient deleters are the Vasa-Cre mice they are using (more about mice later) by crossing them with some other fl/fl target strain.*

Data generated from analysis of this *Lsd1* conditonal allele in other tissues (submitted elsewhere) suggests that LSD1 protein (or RNA) can be extremely stable. For example, in some cells, LSD1 is still robustly detected 8 weeks after complete deletion of the floxed allele. Also in our hands, crossing *Cre*-drivers with different floxed alleles (or in mice from different strains) can lead to different patterns/timing of deletion, often not matching published patterns. For further explanation please see comments below.

*In addition, it would be interesting to determine the level of LSD1 in maternally deleted eggs. Is it the same as in heterozygous females or significantly lower? This information would be very important in evaluating the results of behavioural tests.*

The controls for the immunohistochemistry on *Lsd1*_Vasa_ oocyes in Figure 1 are heterozygotes, suggesting that the “reduced” expression that we quantified is significantly lower than heterozygotes. The Figure 1 legend has been amended to indicate this.

*7) In the first paragraph of the subsection “Loss of maternal LSD1 results in 1-2 cell embryonic arrest”, is the timing of expression of paternal allele known? Please provide experimental evidence or citation.*

Answering this question would require that the maternally mutant embryos be fertilized and develop toward the blastocyst stage, or require RNA analysis of hybrid crosses that contain SNPs to distinguish alleles. However as we show in the manuscript, the maternal mutants do not develop when fertilized by wild-type males. In the occasional embryo that cleaves to the 2C stage, we do not detect Lsd1 transcripts (now shown in Figure 3—figure supplement 3), but this could be due to the lack of full ZGA and may not be due to the lack of transcription from the paternal allele. We have generated RNA from 2C embryos derived from hybrid crosses (CAST/B6), but the *Lsd1* allele does not have SNPs, so allele specific expression could not be interrogated.

*8) In the first paragraph of the subsection “Loss of maternal LSD1 results in 1-2 cell embryonic arrest”, the development of Lsd1^Gdf9^ control embryos is very poor. Another example how unrelated/uncontrolled influence can significantly affect experimental results. In view of these results I would suggest deleting all results in which Gdf9-Cre is used. In the last paragraph of the aforementioned subsection, development of embryos from Zp3-Cre crosses is obviously much better than from Gdf9 crosses, strain effect? Another reason to delete Gdf9 results as those two groups should be comparable. 3- and 4-cell stage embryos appear very unhealthy.*

Some of the slight phenotypic differences between *Lsd1*_Zp3_ and *Lsd1*_Gdf9_ mutants may be due to the fact that the experiments were performed in two difference laboratories (Macfarlan and Katz). As a result, there may be subtle differences in developmental timing, strain background, etc. Nevertheless, we believe that the small differences are real and wish to report the result. To make the manuscript more clear, we have moved the *Lsd1*_Gdf9_ results to a supplemental figure and modified the text.

*9) One note of caution on the behavioural effects is that there could be a potential confound of litter effect, given that factors such as litter size in the early postnatal period could have a programming effect on a variety of adult behaviours – and so could be quite independent of history of Lsd1 deficiency per se – and there is a strong effect on litter size (Figure 4). Stronger data on the effect on imprinted genes (see above) would help negate a possible confounding effect such as this.*

Despite the strong effect on litter size in *Lsd1*_Vasa_ mutants, hypomorphic animals were obtained from litters ranging between 1-4 animals. Nevertheless, the behavior effects were observed in 100% of the animals, regardless of the size of the litter to which they were born. This suggests that the effect is independent of litter size. Text has been added to the manuscript to make this point.

10) From the RNA-seq datasets, the authors comment on mis-regulation of interesting DNA methylation modulators, such as Dnmt1, Tet1, Trim28, Zfp57, Dppa3/Stella in 2-cell embryos from Lsd1^Zp3-^Cre females. Ideally, these effects would be validated by RT-qPCR. Also, this mis-regulation could be more apparent than real. For example, a two-fold up-regulation of Dnmt1 could represent the same number of oocyte-derived Dnmt1 transcripts, but a reduction in the total mRNA pool in the 2-C embryo owing to impaired zygotic genome activation, so it is a relative rather than absolute change.

We agree. We have performed RT-PCR and added it to the manuscript (Figure 3—figure supplement 4). The RT-PCR data is consistent with the reported RNA-seq data.

[Editors' note: further revisions were requested prior to acceptance, as described below.]

Reviewer #1: The authors addressed (or attempted to address) most of the comments. They also provided some additional experiments and discussed why some of the requested experiments could not be performed. The arguments (shortage of time, being out of the scope of the investigation, limit imposed by the available number of embryos) seem overall reasonable and justified. The manuscript is of sufficient quality and interest to justify publication in eLife though some additions in the text are necessary.

Additional comment: Discussion is mostly concentrating on relatively trivial and unexplained results dealing with imprinting errors observed in two dead pups. The authors tried to explain the reason for using dead pups and how they controlled for possible artefacts. This explanation is not entirely satisfactory and possible technical problems could explain why two samples significantly differ. I certainly do not see the need to provide so much space in illustrations, results and discussion to this subject. The rest of Discussion is mostly repetition of the Results. There is no attempt to speculate about some puzzling observations. Presence of small amount of protein in supposedly Vasa-Cre mediated deleted oocytes has been partially explained in the authors' reply to reviewers so why not add those thought to the discussion. Suggestion that possible reason may be extreme stability of LSD1 protein or RNA is not logical. If this would be the case one would expect to see even more protein or RNA in the oocytes in which deletion happens much later when mediated by Gdf9-Cre and ZP3-Cre.

*The authors (again in the accompanying letter) state that differences between the Gdf9-Cre and ZP3-Cre experiments may be caused by different locations. They could have easily check if this is true. As it stands there is very significant difference and readers may draw all kinds of conclusions from what could be trivial fact that one lab is better in growing embryos or handling mice. However, it is also possible that the difference is caused by earlier action of GDF9 which would I turn deprive oocytes of histone demethylase few days earlier. However if LSD1 protein or RNA is so long lived this explanation is useless. One would really like to see those puzzles discussed a bit better. Reviewer #2: As before, this manuscript on the consequences of ablation of LSD1/KDM1A in mouse oocytes has three components: phenotypic and molecular characterisation of defects in oocytes and preimplantation embryos; description of behavioural effects in rare surviving adult progeny obtained after early maternal germline ablation of LSD1 (Vasa-Cre); and molecular analysis of imprinted genes in pups from such maternal germline mutants.*

Some connections can be made between these phenotypic outcomes, but they can be considered separate facets of the story. Thus, the reported behavioural defects could be completely unrelated to altered imprinting, but the analysis of imprinted genes offers precedent that maternal lack of LSD1 could give rise to early epigenetic errors that persist into adult life. In the revised manuscript, the characterisation of the molecular defects in early embryos is significantly improved with the addition of the RNA-seq data from Lsd1:Zp3-Cre oocytes. These added data allow the authors to conclude more strongly that there is a defect in zygotic genome activation, since the disruption in gene expression in oocytes is far less extensive. However, whether the ZGA defect depends upon activity of LSD1 in the zygote/embryo or other factors stored in the oocyte that depend upon LSD1 in the oocyte cannot be distinguished. The manuscript from Ancellin et al., addresses this point. Nevertheless, the informatic analysis of the RNA-seq datasets looks very robust. Regarding the other two main aspects of the manuscript, I retain some concerns. I think the authors could have gone about the expression and methylation analysis in a more robust way. Given the prominence given to this analysis in the manuscript, I would have considered that sacrificing a couple of pups for more refined molecular analysis, rather than relying on the approach adopted of extracting DNA and RNA from histological sections of deceased pups, would have been appropriate. It would have given this analysis far greater credibility: we would have had the confidence that pristine, freshly isolated and defined tissues were being analysed, rather than an aggregation of tissue of unknown quality from a sagittal section. On a side point, regarding the result for Grb10 (Figure 6—figure supplement 1), it is not clear that the authors are actually looking at the germline DMR, as this region should be fully unmethylated on the paternal allele of control animals, whereas both alleles seem to be fully methylated. (The germline DMR is unmethylated on the paternal allele in all tissues and, as far as I am aware, tissue-specific expression status of Grb10 occurs without changes in the methylation of the germline DMR.) Regarding the behavioural tests, I am not sure the issue of litter size and behavioural effects is fully dealt with. Were the litter sizes similar for the various control groups, given the very small litter sizes of the maternal mutants? The definitive ways to get around possible confound of litter size when there seems to be such a profound effect on litter size would be to reduce the litter size of the controls groups to those of the maternal mutant class, or undertake cross-fostering to ensure control and maternal mutant groups all experience similar postnatal care. In as much as it is not possible to expect the authors to undertake the study in this way now, I do strongly urge them to include explicit statements in the Materials and methods on the postnatal history of the adults, both maternal mutant and control groups, relating to litter size (exact litter sizes for all groups). There remains a potential confounding factor here, and readers need to be able to appreciate it.

We thank the reviewers for their helpful suggestions. We agree with the points that they have raised in the new round of reviews and have made all of their suggested changes and additions to the text (largely in the Discussion). In addition, we have removed the confusing *Grb10* imprinting data from Figure 6—figure supplement 1 and the corresponding text in the results. Finally, as requested, we have added a new table and text describing the table in the Methods. This new table lists the litter sizes of all perinatal lethal animals and all animals assayed in the behavioral assays.